# Universal Sparse Autoencoders: Interpretable Cross-Model Concept Alignment

**Harrish Thasarathan** [1 2]   **Julian Forsyth** [1]   **Thomas Fel** [3]   **Matthew Kowal** [1 2 4 5]   **Konstantinos G. Derpanis** [1 6 2 7]

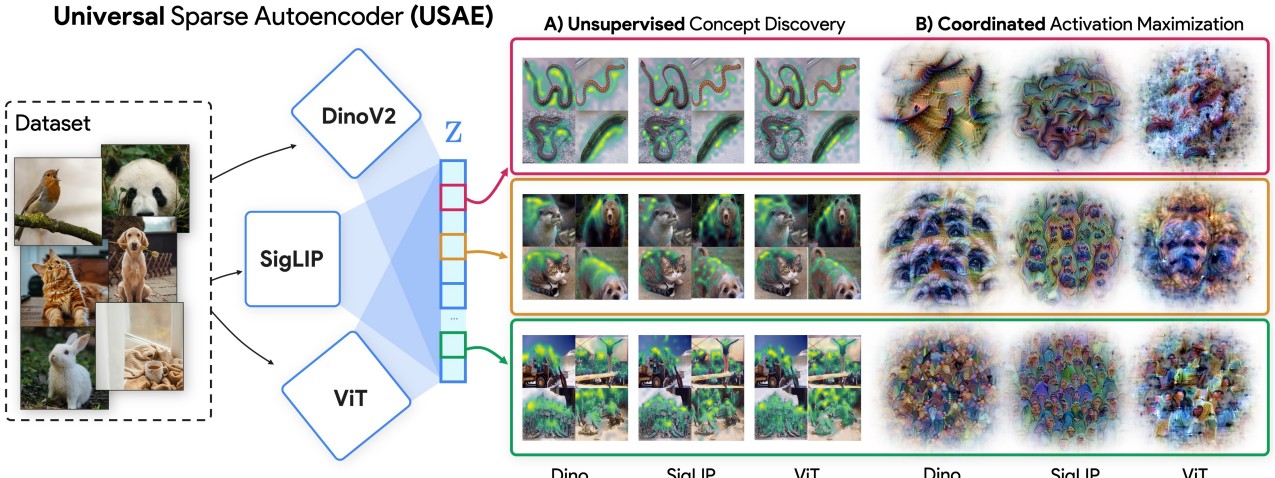

*Figure 1.* **Overview of Universal Sparse Autoencoders.** (A) We introduce *Universal Sparse Autoencoders* (USAEs), a method for discovering common concepts across multiple different deep neural networks. USAEs are simultaneously trained on the activations of multiple models and are constrained to share an aligned and interpretable dictionary of discovered concepts. (B) We also demonstrate one immediate application of USAEs, *Coordinated Activation Maximization*, where optimizing the inputs of multiple models to activate the same concepts reveals how different models encode the same concept. Visualization reveals interesting concepts at various levels of abstraction, such as 'curves' (top), 'animal haunch' (middle) and 'the faces of crowds' (bottom). Code: github.com/YorkUCVIL/UniversalSAE.

## Abstract

We present *Universal Sparse Autoencoders* (US-AEs), a framework for uncovering and aligning interpretable concepts spanning multiple pretrained deep neural networks. Unlike existing concept-based interpretability methods, which focus on a single model, USAEs jointly learn a universal concept space that can reconstruct and interpret the internal activations of multiple models at once. Our core insight is to train a single, overcomplete sparse autoencoder (SAE) that ingests activations from any model and decodes them to approximate the activations of any other model under consideration. By optimizing a shared ob-jective, the learned dictionary captures common factors of variation—*concepts*—across different tasks, architectures, and datasets. We show that USAEs discover *semantically coherent* and *important* universal concepts across vision models; ranging from low-level features (e.g., colors and textures) to higher-level structures (e.g., parts and objects). Overall, USAEs provide a powerful new method for interpretable cross-model analysis and offers novel applications—such as coordinated activation maximization—that open avenues for deeper insights in multi-model AI systems.

## 1. Introduction

In this work, we focus on discovering interpretable concepts shared among multiple pretrained deep neural networks (DNNs). The goal is to learn a *universal concept space* – a joint space of concepts – that provides a unified lens into the hidden representations of diverse models. We define concepts as the abstractions each network captures that transcend individual data points—spanning low-level features (e.g., colors and textures) to high-level attributes (e.g.,

[1]York University, Toronto, Canada [2]Vector Institute, Toronto, Canada [3]Kempner Institute, Harvard University, Boston, USA [4]FAR.AI [5]Trajectory Labs, Toronto [6]University of Toronto, Toronto, Canada [7]Samsung AI Centre, Toronto. Correspondence to: Harrish Thasarathan <harryt@yorku.ca>.

*Proceedings of the 42$^{nd}$ International Conference on Machine Learning*, Vancouver, Canada. PMLR 267, 2025. Copyright 2025 by the author(s).

emotions like *horror* and ideas like *holidays*).

Grasping the underlying representations within DNNs is crucial for mitigating risks during deployment (Buolamwini & Gebru, 2018; Hansson et al., 2021), fostering the development of innovative model architectures (Kowal et al., 2025; Darcet et al., 2024), and abiding by regulatory frameworks (Commision, 2021; House, 2023). Prior interpretability efforts often center on dissecting a single model for a specific task, leaving risk management unmanageable when each network is analyzed in isolation. With a growing number of capable DNNs, finding a canonical basis for understanding model internals may yield more tractable strategies for managing potential risks.

Recent work supports this possibility. The core idea behind 'foundation models' (Henderson et al., 2023) presupposes that any DNN trained on a large enough dataset should encode concepts that generalize to an array of downstream tasks for that modality. Moreover, recent work (Moschella et al., 2022) has shown that regardless of architecture, initialization, and task, differently trained models may yield semantically equivalent latent representations, and recent studies (Dravid et al., 2023; Kowal et al., 2024a) even found shared concepts across vision models. This implies that universality may be more widespread than previously assumed. However, current techniques for identifying universal features (Dravid et al., 2023; Huh et al., 2024; Kowal et al., 2024a) typically operate *post-hoc*, extracting concepts from individual models and then matching them through compute-intensive filtering or optimization. This approach is limited in scalability, lacks the efficiencies of gradient-based training, and precludes *translation* between models within a unified concept space. Consequently, tasks that require simultaneous interaction across multiple models, e.g., *coordinated activation maximization* shown later, become more cumbersome.

To overcome these challenges, we introduce a *universal sparse autoencoder* (USAE), Fig. 1, designed to jointly encode and reconstruct activations from multiple DNNs. Through qualitative and quantitative evaluations, we show that the resulting concept space captures interpretable features shared across all models. Crucially, a USAE imposes concept alignment during its end-to-end training, differing from conventional post-hoc methods. We apply USAEs to three diverse vision models and make several interesting findings about shared concepts: (i) We discover a *broad range of universal concepts*, at low and high levels of abstraction. (ii) We observe a strong correlation between concept *universality* and *importance*. (iii) We provide quantitative and qualitative evidence that DinoV2 (Oquab et al., 2023) admits *unique features* compared to other considered vision models. (iv) Universal training admits shared representations *not uncovered* in model-specific SAE training.

**Contributions.** Our main contributions are as follows. First, we introduce USAEs: a framework that learns a shared, interpretable concept space spanning multiple models, with focus on visual tasks. Second, we present a detailed analysis contrasting universal concepts against model-specific concepts, offering new insights into how large vision models—trained on diverse tasks and datasets—compare and diverge in their internal representations. Finally, we demonstrate a novel USAE application, *coordinated activation maximization*, showcasing simultaneous visualization of universal concepts across models.

## 2. Related work

Our work introduces a novel *concept-based interpretability* method that adapts *SAEs* to discover *universal concepts*. We now review the most relevant works in each of these fields.

**Concept-based interpretability** (Kim et al., 2018) emerged as a response to the limitations of attribution methods (Simonyan et al., 2014; Zeiler & Fergus, 2014; Bach et al., 2015; Springenberg et al., 2014; Smilkov et al., 2017; Sundararajan et al., 2017; Selvaraju et al., 2017; Fong et al., 2019; Fel et al., 2021; Muzellec et al., 2024), which, despite being widely used for explaining model predictions, often fail to provide a structured or human-interpretable understanding of internal model computations (Hase & Bansal, 2020; Hsieh et al., 2021; Nguyen et al., 2021; Colin et al., 2021; Kim et al., 2022; Sixt et al., 2020). Attribution methods highlight input regions responsible for a given prediction, the *where*, but do not explain *what* the model has learned at a higher level. In contrast, concept-based approaches aim to decompose internal representations into human-understandable *concepts* (Genone & Lombrozo, 2012). The main components of concept-based interpretability approaches can generally be broken down into two parts (Fel et al., 2023b): (*i*) concept discovery, which extracts and visualizes the interpretable units of computation and (*ii*) concept importance estimation, which quantifies the importance of these units to the model output. Early work explored 'closed-world' concept settings in which they evaluated the existence of pre-defined concepts in model neurons (Bau et al., 2017) or layer activations (Kim et al., 2018). When access to an aligned text-image representation space is available, output-level image representations can be decomposed into interpretable components using text representations as a dictionary (Gandelsman et al., 2024). Similar to our work, 'open-world' concept discovery methods do not assume the set of concepts is known a priori. These methods pass data through the model and cluster the activations to discover concepts and then apply a concept importance method on these discoveries (Ghorbani et al., 2019; Zhang et al., 2021; Fel et al., 2023c; Graziani et al., 2023; Vielhaben et al., 2023; Kowal et al., 2024a;b).

**Sparse Autoencoders** (SAEs) (Cunningham et al., 2023; Bricken et al., 2023; Rajamanoharan et al., 2024; Gao et al., 2024; Menon et al., 2024) are a specific instance of dictionary learning (Rubinstein et al., 2010; Elad, 2010; Tošić & Frossard, 2011; Mairal et al., 2014; Dumitrescu & Irofti, 2018) that has regained attention (Chen et al., 2021; Tasissa et al., 2023; Baccouche et al., 2012; Tariyal et al., 2016; Papyan et al., 2017; Mahdizadehaghdam et al., 2019; Yu et al., 2023) for its ability to uncover interpretable concepts in DNN activations. This resurgence stems from evidence that individual neurons are often *polysemantic*—i.e., they activate for multiple, seemingly unrelated concepts (Nguyen et al., 2019; Elhage et al., 2022)—suggesting that deep networks encode information in *superposition* (Elhage et al., 2022). SAEs tackle this by learning a sparse (Hurley & Rickard, 2009; Eamaz et al., 2022) and *overcomplete* representation, where the number of concepts exceeds the latent dimensions of the activation space, encouraging disentanglement and interpretability. While SAEs and clustering bear mathematical resemblance, SAEs benefit from gradient-based optimization, enabling greater scalability and efficiency in learning structured concepts. Though widely applied in natural language processing (NLP) (Wattenberg & Viégas, 2024; Clarke et al., 2024; Chanin et al., 2024; Tamkin et al., 2023), SAEs have also been used in vision (Fel et al., 2023b; Surkov et al., 2024; Bhalla et al., 2024a). Early work compared SAEs to clustering and analyzed early layers of Inception v1 (Mordvintsev et al., 2015; Gorton, 2024), revealing hypothesized but hidden features. More recently, SAEs have been leveraged to construct text-based concept bottleneck models (Koh et al., 2020) from CLIP representations (Radford et al., 2021; Rao et al., 2024; Parekh et al., 2024; Bhalla et al., 2024b), showcasing their versatility across modalities. Unlike prior work that apply SAEs independently to models, here we consider a joint application of SAEs fit simultaneously across diverse models.

**Feature Universality** studies the shared information across different DNNs. One approach, Representational alignment, quantifies the mutual information between different sets of representations—whether across models or between biological and artificial systems (Kriegeskorte et al., 2008; Sucholutsky et al., 2023). Typically, these methods rely on paired data (e.g., text-image pairs) to compare encodings across modalities. Recent work suggests that foundation models, regardless of their training modality, may be converging toward a shared, *Platonic* representation of the world (Huh et al., 2024). Another line of research focuses on identifying universal features across models trained on different tasks. Rosetta *Neurons* (Dravid et al., 2023) identify image regions with correlated activations across models, while Rosetta *Concepts* (Kowal et al., 2024a) extract concept vectors from video transformers by analyzing shared exemplars. These methods perform post-hoc mining of uni-

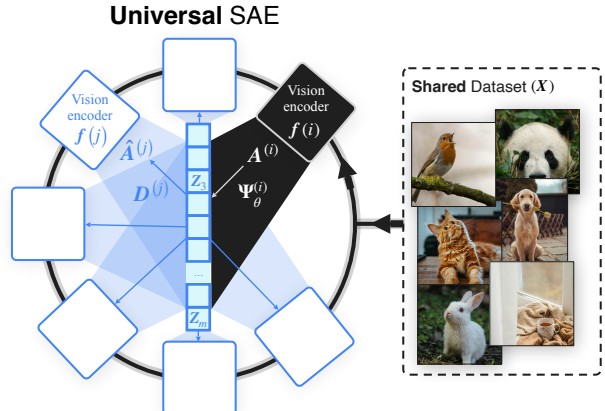

**Universal** SAE

*Figure 2.* **USAE training process.** In each forward pass during training, an encoder of model $i$ is randomly selected to encode a batch of that model's activations, $\boldsymbol{Z} = \boldsymbol{\Psi}_\theta^{(i)}(\boldsymbol{A}^{(i)})$. The concept space, $\boldsymbol{Z}$, is then decoded to reconstruct every model's activations, $\hat{\boldsymbol{A}}^{(j)}$, using their respective decoders, $\boldsymbol{D}^{(j)}$.

versal concepts rather than learning a shared conceptual space. This reliance on retrospective discovery is computationally prohibitive for many models and prevents direct concept translation between architectures. A concurrent study (Lindsey et al., 2024) explores training SAEs (termed *crosscoders*) between different states of the same model before and after fine-tuning. In contrast, our work discovers universal concepts shared *across* distinct model architectures for vision tasks.

## 3. Method

**Notations.** Let $\|\cdot\|_2$ and $\|\cdot\|_F$ denote the $\ell_2$ and Frobenius norms, respectively, and set $[n] = \{1, \ldots, n\}$. We focus on a broad representation learning paradigm, where a DNN, $\boldsymbol{f} : \mathcal{X} \to \mathcal{A}$, maps data from $\mathcal{X}$ into a feature space, $\mathcal{A} \subseteq \mathbb{R}^d$. Given a dataset, $\boldsymbol{X} \subseteq \mathcal{X}$ of size $n$, these activations are collated into a matrix $\boldsymbol{A} \in \mathbb{R}^{n \times d}$. Each row $\boldsymbol{A}_i$ (for $i \in [n]$) corresponds to the feature vector of the $i$-th sample.

**Background.** The main goal of a Sparse Autoencoder (SAE) is to find a sparse re-interpretation of the feature representations. Concretely, given a set of $n$ inputs, $\boldsymbol{X}$ (e.g., images or text) and their encoding, $\boldsymbol{A} = \boldsymbol{f}(\boldsymbol{X}) \in \mathbb{R}^{n \times d}$, an SAE learns an encoder $\boldsymbol{\Psi}_\theta(\cdot)$ that maps $\boldsymbol{A}$ to *codes* $\boldsymbol{Z} = \boldsymbol{\Psi}_\theta(\boldsymbol{A}) \in \mathbb{R}^{n \times m}$, forming a sparse representation. This sparse representation must still allow faithful reconstruction of $\boldsymbol{A}$ through a learned *dictionary* (decoder) $\boldsymbol{D} \in \mathbb{R}^{m \times d}$, i.e., $\boldsymbol{Z}\boldsymbol{D}$ must be close to $\boldsymbol{A}$. If $m > d$, we say $\boldsymbol{D}$ is *overcomplete*. In this work, we specifically consider an (overcomplete) TopK SAE (Gao et al., 2024), defined as

$$\boldsymbol{Z} = \boldsymbol{\Psi}_\theta(\boldsymbol{A}) = \text{TopK}(\boldsymbol{W}_{\text{enc}}(\boldsymbol{A} - \boldsymbol{b}_{\text{pre}})), \hat{\boldsymbol{A}} = \boldsymbol{Z}\boldsymbol{D}, \quad (1)$$

where $\boldsymbol{W}_{\text{enc}} \in \mathbb{R}^{m \times d}$ and $\boldsymbol{b}_{\text{pre}} \in \mathbb{R}^d$ are learnable weights. The $\text{TopK}(\cdot)$ operator enforces $\|\boldsymbol{Z}_i\|_0 \leq K$ for all $i \in [m]$. The final training loss is given by the Frobenius norm of the

reconstruction error:

$$\mathcal{L}_{\text{SAE}} = \|\boldsymbol{f}(\boldsymbol{X}) - \boldsymbol{\Psi}_\theta\big(\boldsymbol{f}(\boldsymbol{X})\big)\boldsymbol{D}\|_F = \|\boldsymbol{A} - \boldsymbol{Z}\boldsymbol{D}\|_F, \quad (2)$$

with the $K$-sparsity constraint applied to the rows of $\boldsymbol{Z}$.

### 3.1. Universal Sparse Autoencoders (USAEs)

Contrasting standard SAEs, which reinterpret the internal representations of a *single* model, *universal* sparse autoencoders (USAEs) extend this notion across $M$ different models, each with its own feature dimension, $d_i$ (see Fig. 2). Concretely, for model $i \in [M]$, let $\boldsymbol{A}^{(i)} \in \mathbb{R}^{n \times d_i}$ denote the matrix of activations for $n$ samples. The key insight of USAEs is to learn a shared sparse code, $\boldsymbol{Z} \in \mathbb{R}^{n \times m}$, which allows every model to be reconstructed from the same sparse embedding. Specifically, each activation from model $i$ in $\boldsymbol{A}^{(i)}$ is encoded via a model-specific encoder $\boldsymbol{\Psi}_\theta^{(i)}$, as

$$\boldsymbol{Z} = \boldsymbol{\Psi}_\theta^{(i)}(\boldsymbol{A}^{(i)}) = \text{TopK}\big(\boldsymbol{W}_{\text{enc}}^{(i)}(\boldsymbol{A}^{(i)} - \boldsymbol{b}_{\text{pre}}^{(i)})\big). \quad (3)$$

Crucially, once encoded into $\boldsymbol{Z}$, each row of any model $j \in [M]$ can be reconstructed by a model-specific dictionary, $\boldsymbol{D}^{(j)} \in \mathbb{R}^{d_j \times m}$, as

$$\widehat{\boldsymbol{A}}^{(j)} = \boldsymbol{Z}\boldsymbol{D}^{(j)}. \quad (4)$$

By jointly learning all encoder-decoder pairs, $\{(\boldsymbol{\Psi}_\theta^{(i)}, \boldsymbol{D}^{(i)})\}_{i=1}^M$, the USAE enforces a unified concept space, $\boldsymbol{Z}$, that aligns the internal representations of all $M$ models. This shared code not only promotes consistency and interpretability across model architectures, but also ensures each model's features can be faithfully recovered from a *common* set of sparse 'concepts'.

### 3.2. Training USAEs

Recall that $\boldsymbol{X} \subseteq \mathcal{X}$ is our dataset of size $n$, mapped into their respective feature space using DNNs $\boldsymbol{f}^{(1)}, \ldots, \boldsymbol{f}^{(M)}$. A naive approach to train our respective encoder and decoder would simultaneously encode and decode the features of all $M$ models, which quickly grows expensive in memory and computation.

Conversely, randomly sampling a pair of models to encode and decode results in slow convergence. To balance these concerns, we adopt an intermediate strategy (pseudocode detailed in Figure 3) that updates a single encoder-decoder pair at each iteration with a reconstruction loss computed through *all* decoders. Concretely, at each mini-batch iteration, a single model $i \in [M]$ is selected at random, and a batch of features, $\boldsymbol{A}^{(i)} \in \mathbb{R}^{n \times d_i}$, is sampled and encoded into the shared code space, $\boldsymbol{Z} = \boldsymbol{\Psi}_\theta^{(i)}(\boldsymbol{A}^{(i)})$. This code space, $\boldsymbol{Z}$, is then used to reconstruct the feature representation $\boldsymbol{A}^{(j)}$ of every model $j \in [M]$ via its decoder: $\widehat{\boldsymbol{A}}^{(j)} = \boldsymbol{Z}\boldsymbol{D}^{(j)}$, where $\boldsymbol{D}^{(j)}$ is the model-$j$ decoder. All reconstructions are aggregated to form the total loss:

```python
def train_usae(Ψ_θ, D, A, T, Optimizers):
    M = len(Ψ_θ)
    for t in range(T):
        i = random(M)
        Z = Ψ_θ^(i)(A^(i))
        L = 0.0
        for j in range(M):
            Â^(j) = Z @ D^(j)
            L += (A^(j) - Â^(j)).norm(p='fro')
        L.backward()
        Optimizers[i].step()
    return Ψ_θ, D
```

*Figure 3.* **Training Universal Sparse Autoencoder.** During each training iteration, $\mathcal{L}_{\text{Universal}}$ is the aggregated error computed from decoding each activation $\widehat{A}^{(j)}$. We then take an optimizer step for randomly selected encoder $\boldsymbol{\Psi}_\theta^{(i)}$ and associated dictionary $\boldsymbol{D}^{(i)}$.

$$\mathcal{L}_{\text{Universal}} = \sum_{j=1}^M \|\boldsymbol{A}^{(j)} - \widehat{\boldsymbol{A}}^{(j)}\|_F \quad (5)$$

$$= \sum_{j=1}^M \|\boldsymbol{A}^{(j)} - \boldsymbol{\Psi}_\theta(\boldsymbol{A}^{(i)})\boldsymbol{D}^{(j)}\|_F. \quad (6)$$

Using this universal loss, backpropagation updates the chosen encoder $\boldsymbol{\Psi}_\theta^{(i)}$ and decoder $\boldsymbol{D}^{(i)}$. This method promotes concept alignment, ensures an equal number of updates between encoders and decoders, and strikes a practical balance between training speed and memory usage.

### 3.3. Application: Coordinated Activation Maximization

A common technique for interpreting individual neurons or latent dimensions in deep networks is *Activation Maximization (AM)* (Olah et al., 2017; Tsipras et al., 2019; Santurkar et al., 2019; Engstrom et al., 2019; Ghiasi et al., 2021; 2022; Fel et al., 2023a; Hamblin et al., 2024). AM involves synthesizing an input that maximally activates a specific component of a model—such as a neuron, channel, or concept vector (Williams, 1986; Mahendran & Vedaldi, 2015; Kim et al., 2018; Fel et al., 2023c). However, in the case of a USAE, the learned latent space is explicitly structured to capture *shared concepts* across multiple models. This shared space enables a novel extension of AM: *Coordinated Activation Maximization*, where a common concept index, $k$, is simultaneously maximized across all aligned models.

Given $M$ models, our objective is to optimize one input per model, $\boldsymbol{x}_\star^{(1)}, \ldots, \boldsymbol{x}_\star^{(M)}$, ensuring that all inputs maximally activate the same concept dimension $k$. This approach enables the visualization of how a single concept manifests across different models. By comparing these optimized inputs, we can identify both *consistent* and *divergent* representations of the same underlying concept. Let $\boldsymbol{x}^{(i)}$ denote

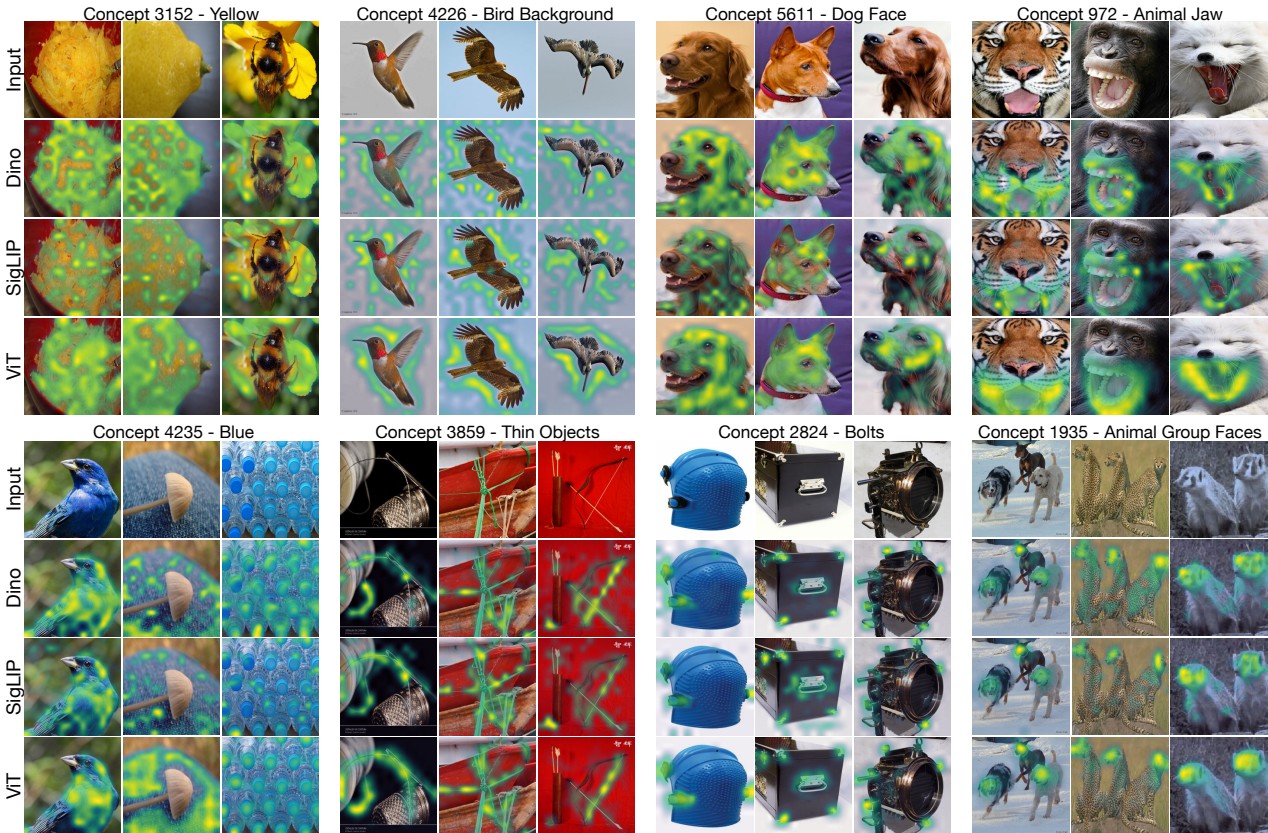

*Figure 4.* **Qualitative results of universal concepts.** We discover and visualize heatmaps of universal concepts across a broad range of visual abstractions, where bright green denotes a stronger activation of a given concept. We observe colors, basic shapes, foreground-background, parts, objects and their groupings across *all considered models*.

the input to model $i$, and let $\boldsymbol{f}^{(i)}(\boldsymbol{x}^{(i)}) \in \mathbb{R}^{d_i}$ represent its internal activations. Each model is associated with a USAE encoder $\boldsymbol{\Psi}_\theta^{(i)}$, which maps activations to the shared concept space. The activation of concept $k$ for model $i$ given input $\boldsymbol{x}^{(i)}$ is defined as

$$\boldsymbol{Z}_k^{(i)}(\boldsymbol{x}) = \left[ \boldsymbol{\Psi}_\theta^{(i)} \left( \boldsymbol{f}^{(i)}(\boldsymbol{x}) \right) \right]_k , \qquad (7)$$

where $k$ indexes the universal concept dimension in the USAE. The goal is to independently optimize each $\boldsymbol{x}^{(i)}$ such that it maximizes the activation of the same concept $k$ across all $M$ models:

$$\boldsymbol{x}_\star^{(i)} = \underset{\boldsymbol{x} \in \mathcal{X}}{\arg\max} \, \boldsymbol{Z}_k^{(i)}(\boldsymbol{x}^{(i)}) - \lambda \mathcal{R}(\boldsymbol{x}^{(i)}), \qquad (8)$$

where $\mathcal{R}(\boldsymbol{x})$ is a regularizer that promotes natural and interpretable inputs (e.g., total variation, $\ell_2$ penalty, or data priors), and $\lambda$ controls its strength. In all experiments, we follow the optimization and regularization strategy of Maco (Fel et al., 2023a), which optimizes the input phase while preserving its magnitude. Once the optimized inputs $\boldsymbol{x}_\star^{(i)}$ are obtained for each model, they reveal the specific structures or features (e.g., model- or task-specific biases) that model $i$ associates with this universal concept.

## 4. Experimental Results

This section is split into six parts. We first provide experimental implementation details. Then, we qualitatively analyze universal concepts discovered by USAEs (Sec. 4.1). Next, we provide a quantitative analysis of USAEs through the validation of activation reconstruction (Sec. 4.2), measuring the universality and importance of concepts (Secs. 4.3), and investigating the consistency between concepts in US-AEs and individually trained SAE counterparts (Sec. 4.4). Finally, we provide a finer-grained analysis via the application of USAEs to *coordinated activation maximization* (Sec. 4.5).

**Implementation Details.** We train a USAE on the final layer activations of three popular vision models: Di-noV2 (Oquab et al., 2023; Darcet et al., 2024), SigLIP (Zhai et al., 2023), and ViT (Dosovitskiy et al., 2020) (trained on ImageNet (Deng et al., 2009)). These models, sourced from the `timm` library (Wightman, 2019), were selected due to their diverse training paradigms—image and patch-level discriminative learning (DinoV2), image-text contrastive learning (SigLIP), and supervised classification (ViT). For all experiments, we train the USAE on the ImageNet training set, while the validation set is reserved for qualitative visualizations and quantitative evaluations. Our USAE is

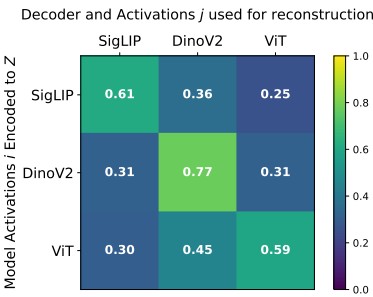

*Figure 5.* **Cross model activation reconstruction.** Each entry $(i, j)$ represents the average $R^2$ score when activations $\boldsymbol{A}^{(i)}$ from model $i$ are encoded into the shared code space, $\boldsymbol{Z}$, then decoded via $\boldsymbol{D}^{(j)}$ to reconstruct $\widehat{\boldsymbol{A}}^{(j)}$. Positive off-diagonal $R^2$ scores indicate the presence of shared features across models captured by USAEs.

trained on the final layer representations of each vision model, as previous work showed final-layer features facilitate improved concept extraction and yield accurate estimates of feature importance (Fel et al., 2023b). We base our SAE off of the TopK Sparse Autoencoder (SAE) (Gao et al., 2024) and for all experiments, use a dictionary of size 6144. We train all USAEs on a single Nvidia RTX 6000 GPU, with training completing in approximately three days (see Appendix A.1 for more implementation details).

### 4.1. Universal Concept Visualizations

We qualitatively validate the most important universal concepts found by USAEs. We determine concept importance by measuring its relative energy towards reconstruction (Gillis, 2020), where the energy of a concept $k$ is defined as

$$\text{Energy}(k) = \|\mathbb{E}_{\boldsymbol{x}}[\boldsymbol{Z}_k(\boldsymbol{x})]\boldsymbol{D}_k\|_2^2. \tag{9}$$

This measures how much each concept contributes to reconstructing the original features – formally, the squared $\ell_2$ norm of the average activation of a concept multiplied by its dictionary element. Higher energy concepts have a greater impact on the reconstruction.

Figure 4 presents eight representative concepts selected from the 100 most important USAE concepts. These concepts span a diverse range of ImageNet categories, demonstrating the ability of USAEs to capture meaningful features across multiple levels of abstraction and complexity (Olah et al., 2017; Fel et al., 2024). At lower levels, the USAE extracts fundamental color concepts, such as 'yellow' and 'blue', activating over broad spatial regions across multiple classes. Notably, the blue bottle caps example highlights a precisely captured checkerboard pattern, demonstrating spatial precision. At intermediate levels, the USAE uncovers structural relationships consistent across models, such as foreground-background contrasts (e.g., birds against the sky) and thin, wiry objects, independent of model architecture. At higher levels, it identifies object-part concepts, like 'dog face', excluding eye regions, and 'bolts', which activate across materials like metal and rubber. Finally, the

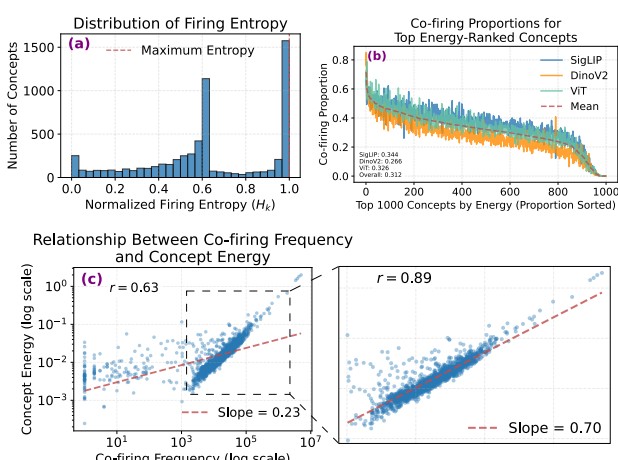

*Figure 6.* **Quantitative analysis of universality and importance of USAE concepts via co-firing rates.** (a) Histogram of firing entropy across all $k$ concepts. We observe a bimodal distribution over firing entropy with peaks at $H_k = 1$ and $H_k = 0.6$, demonstrating a group of concepts that fire uniformly across models and a group that preferentially activates for some models. (b) Proportion of concept co-fires for the top 1000 energetic concepts per model. The first 200 concepts co-fire between $60 - 80\%$ of the time suggesting high universality. (c) Relationship between concept co-firing frequency and concept energy. We show all concepts (left) and only frequently co-firing concepts ($\geq 1000$ co-fires) (right). The correlation strengthens ($r = 0.63$ vs $r = 0.89$) when focusing on high-frequency concepts, suggesting a strong correlation between how energetic a concept is and its universality.

USAE reveals fine-grained, compositional concepts such as 'mouth-open animal jaws' and 'faces of animals in a group', which generalize across ImageNet classes and persist even in ViT, despite its lack of explicit structured supervision.

Overall, these findings show that USAEs discover robust, generalizable concepts that persist across different architectures, training tasks, and datasets. This highlights their ability to reveal invariant, semantically meaningful representations that transcend the specifics of any single model.

### 4.2. Validation of Cross-Model Reconstruction

A viable universal space of concepts should enable the reconstruction of activations from any model. To quantify the reconstruction performance, we use the coefficient of determination, or $R^2$ score (Wright, 1921), which measures the proportion of variance in the original activations that is captured by the reconstructed activations, relative to the mean activation baseline, $\bar{\boldsymbol{A}}$. The $R^2$ score is defined as

$$R^2 = 1 - \|\boldsymbol{A} - \widehat{\boldsymbol{A}}\|_F^2 / \|\boldsymbol{A} - \bar{\boldsymbol{A}}\|_F^2, \tag{10}$$

where $\|\boldsymbol{A} - \widehat{\boldsymbol{A}}\|_F^2$ represents the residual sum of squares (the reconstruction error), and $\|\boldsymbol{A} - \bar{\boldsymbol{A}}\|_F^2$ is the total sum of squares (the variance of the original activations relative to their mean). A higher $R^2$ indicates better reconstruction quality, with a score of one for a perfect reconstruction.

Figure 5 shows the $R^2$ scores as a confusion matrix across all three models. As expected, self-reconstruction along the diagonal achieves the highest explained variance, confirming the USAE's effectiveness when encoding and decoding within the same model. More notably, positive off-diagonal $R^2$ scores indicate successful cross-model reconstruction, suggesting the USAE captures shared, likely universal, features. DinoV2 exhibits the highest self-reconstruction performance, aligning with individual SAE results where its $R^2$ score averages 0.8, compared to 0.7 for SigLIP and ViT. This suggests DinoV2 features are sparser and more decomposable, a trend further supported in Secs. 4.3 and 4.5.

### 4.3. Measuring Concept Universality and Importance

Having established the efficacy of cross-model reconstruction, we now assess concept *universality* using *firing entropy* and *co-firing* metrics. We further examine the relationship between *universality* and *importance* in reconstructing ground truth activations.

Let $\tau$ be a threshold value and $\mathcal{V}$ be the ImageNet validation set of patches. Given data points $\boldsymbol{x} \in \mathcal{V}$, let $\boldsymbol{Z}^{(i)}(\boldsymbol{x}) = \boldsymbol{\Psi}_\theta^{(i)}(\boldsymbol{f}^{(i)}(\boldsymbol{x}))$ denote the sparse code from model $i \in [M]$. We define a concept firing for dimension $k$ when $\boldsymbol{Z}_k^{(i)}(\boldsymbol{x}) > \tau$. A co-fire occurs when a concept fires simultaneously across all models for the same input. Formally, for concept dimension $k$, the set of co-fires is defined as

$$\mathcal{C}_k = \{\boldsymbol{x} \in \mathcal{V} : \min_{i \in [M]} \boldsymbol{Z}_k^{(i)}(\boldsymbol{x}) > \tau\}. \tag{11}$$

Similarly, let $\mathcal{F}_k^{(i)} = \{\boldsymbol{x} \in \mathcal{V} : \boldsymbol{Z}_k^{(i)}(\boldsymbol{x}) > \tau\}$ denote the set of "fires" for model $i$ and concept $k$. We are now ready to introduce our two metrics (*i*) Firing Entropy (**FE**) and (*ii*) Co-Fire Proportion (**CFP**).

**Firing Entropy (FE)** measures, for each concept $k$, the normalized entropy across models, as

$$\mathbf{FE}_k = -\frac{1}{\log M} \sum_{i=1}^M p_k^{(i)} \log p_k^{(i)}, \tag{12}$$

where

$$p_k^{(i)} = |\mathcal{F}_k^{(i)}| / \sum_{j=1}^M |\mathcal{F}_k^{(j)}|. \tag{13}$$

The normalization ensures $\mathbf{FE}_k \in [0, 1]$, with $\mathbf{FE} = 1$ indicating a shared concept with uniform firing across models and low entropy indicating that a concept has a model bias and fires for a single architecture or subset.

Figure 6 (a) shows a histogram of firing entropies across all concept dimensions $K$. Fully universal concepts should have a maximum entropy of one, indicating uniform firing across models. Our results exhibit a bimodal distribution, with over 1000 concepts at peak entropy, confirming the USAE learns a strongly universal concept space. A second group shows moderate entropy, indicating concepts that fa-

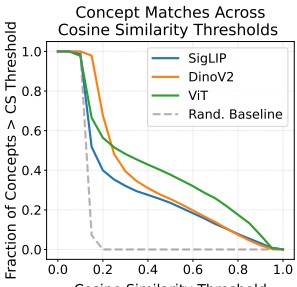

| Model | AUC | % $\boldsymbol{Z} > 0.5$ |
|---|---|---|
| SigLIP | 0.30 | 0.23 |
| DinoV2 | 0.36 | 0.26 |
| ViT | 0.41 | 0.38 |
| Baseline | 0.13 | 0.00 |

*Figure 7.* **Concept consistency between independent SAEs and Universal SAEs.** (left) Our universal training objective discovers concepts that have overlap (i.e., cosine similarity) with those discovered with independent training. Specifically, ViT has noticeably more overlap, suggesting its simpler architecture and training objective may yield activations that naturally encode universal visual concepts. (right) We consider a cosine similarity (CS) $> 0.5$ as a match between concepts in the SAE and USAE learned dictionaries. Across each vision model used in training, the Area Under the Curve (AUC) suggests $23 - 37\%$ of the universal concepts $\boldsymbol{Z}$ discovered by our approach exist in independently trained SAEs.

vor two models but not all three. Few concepts fall in the low-entropy range (0.0–0.2), suggesting most are shared rather than model-specific. Appendix A.2 further examines these low-entropy concepts, revealing DinoV2's unique encoding of *geometric* features as well as SigLIP's encoding of *textual* features.

**Co-Fire Proportion (CFP)** quantifies how often concepts fire together for the same input. While previous results show many concepts fire uniformly across models, they do not reveal how frequently they co-fire on the same tokens. For each model $i$ and concept $k$, we compute the proportion of total fires that are also co-fires:

$$\mathbf{CFP}_k^{(i)} = |\mathcal{C}_k| / |\mathcal{F}_k^{(i)}|. \tag{14}$$

High co-fire proportions indicate concepts that are more universal, i.e., when one model detects the concept, others tend to as well.

Figure 6 (b) shows the **CFP** for the top 1000 concepts per model. The first $\sim$100 concepts exhibit high co-firing $(> 0.5)$, activating together 50–80% of the time, indicating a core set of consistently recognized concepts across networks. The gradual decline in **CFP** suggests a spectrum of universality, from widely shared to model-specific. For our chosen models, we again notice a pattern distinguishing DinoV2, which has a lower co-firing proportion (0.266) compared to SigLIP (0.344) and ViT (0.326), suggesting the latter two share more concepts. This may stem from DinoV2's architecture and distillation-based training, which enhance its adaptability to diverse vision tasks (Amir et al., 2022). These findings also hint at a correlation between co-firing and concept importance, raising the question: How important are these highly co-firing features?

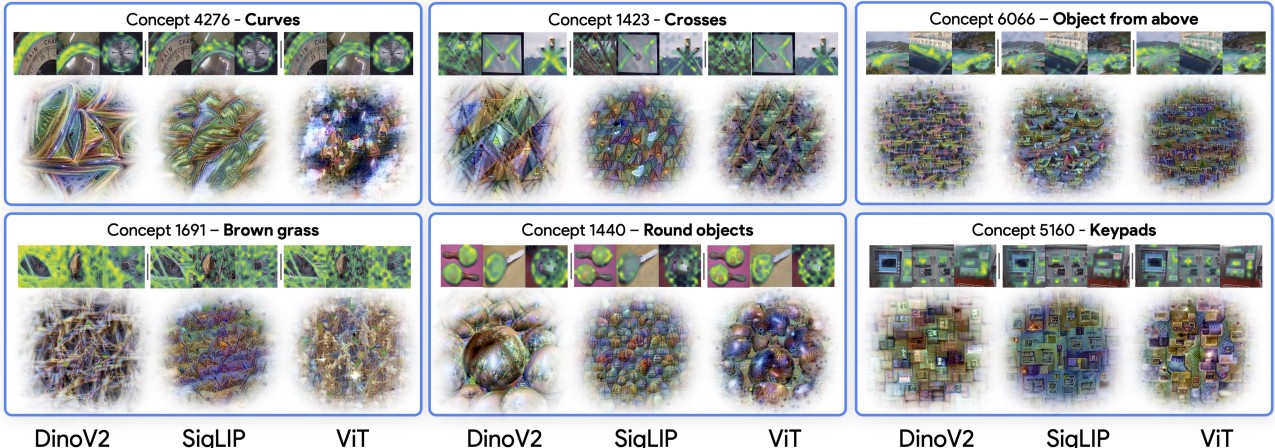

*Figure 8.* **Coordinated Activation Maximization.** We show results for the three model USAE along with dataset exemplars, where bright green denotes stronger activation of the concept. We visualize the maximally activating input for a broad range of concepts, including basic shape compositions, textures, and various objects.

To answer this, we plot the co-fire frequency of all concepts as well as their energy-based importance in Fig. 6 (c). We see a moderate positive correlation $r = 0.63$, slope $= 0.23$; however, zooming into concepts with $> 1000$ co-fires, shows a much stronger correlation. Indeed, past a certain threshold, co-firing frequency becomes highly predictable of concept importance. This suggests that **the most important concept are also highly universal**, firing consistently across models.

### 4.4. Concept Consistency Between USAEs and SAEs

How many concepts discovered under our universal training regime are present in an independently trained SAE for a single model? Further, what percentage of highly universal concepts appear in these same independently trained SAEs? To assess the alignment between independently-trained and universal SAEs, we analyze the similarity of their learned conceptual spaces. We quantify concept overlap by computing pairwise cosine similarities between decoder vectors and use the Hungarian algorithm (Kuhn, 1955) to optimally align concepts, measuring consistency across models.

Figure 7 presents concept consistency distributions across models. For a baseline to compare against, we sample concept vectors from normal distributions, where the mean and variance are those of each independent model's dictionary. We observe that ViT has the strongest concept overlap with 38% of its concepts having a cosine similarity $> 0.5$ with its independent counterpart. This suggests ViT's conceptual representation under the independent SAE objective is most well preserved under universal training. USAEs achieve far better performance than the baseline (Area Under the Curve (AUC)=0.13) across models, suggesting that universal training preserves meaningful concept alignments rather than learn entirely new representations. On the other hand the relatively low proportion of overlap (23% and 26% for SigLIP and DinoV2, respectively) for concepts indicates

that **universal training discovers concepts that may not emerge in independent training**.

When looking at the *top 1,000 co-firing* concepts (see Sec. A.5.1) we find an overall increase in consistency between individual and universal concepts: the most universal (highest co-firing) concepts are more likely to be found in each model's respective independently trained SAE. Universal training naturally selects for concepts that are well-represented across all models, since these will better minimize the total reconstruction loss, biasing towards discovering fundamental visual concepts that all models have learned to represent. Independently trained SAEs have no such selection pressure, learning to represent any concept that helps reconstruction, including architecture or objective specific concepts that are not universal.

### 4.5. Coordinated Activation Maximization

Figure 8 shows a visual comparison of several universal concepts and their corresponding coordinated activation maximization inputs. Our method produces interpretable visualizations for a given USAE dimension across all models for a broad range of visual concepts. We show examples of all models encoding low-level visual primitives, e.g., 'curves' and 'crosses'. Other basic entities are also shown, like 'brown grass' texture and 'round objects'. Finally, we visualize higher-level concepts corresponding to 'objects from above' and 'keypads'. In all cases, our coordinated activation maximization method produces plausible visual phenomenon that can be used to *identify differences between how each model encodes the same concept*.

For example, we note an interesting difference between DinoV2 and the other models: low-mid level concepts (i.e., left two columns) appear at a much *larger scale* than the other models. Further, as shown in Fig. 1, DinoV2 exhibits stronger activation for the 'curves' concept, particularly for larger curves, compared to the other models. Additionally,

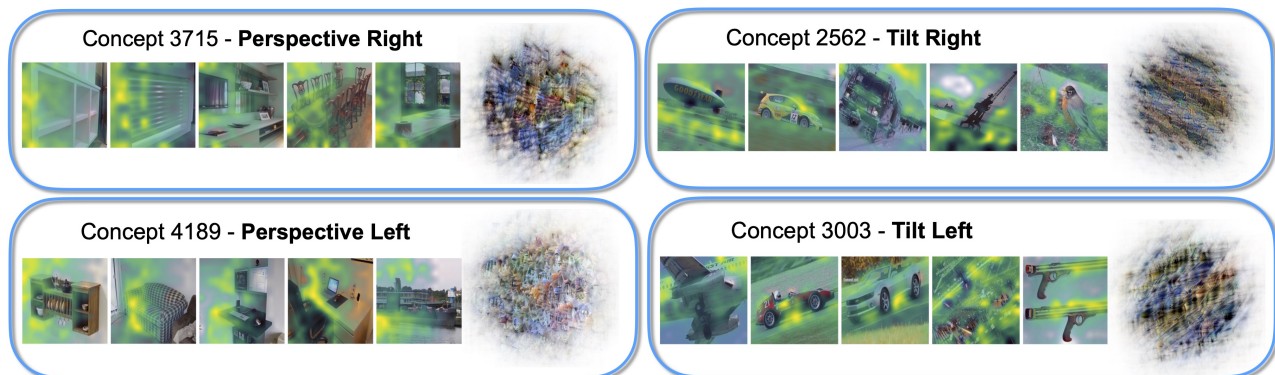

*Figure 9.* **DinoV2 low-entropy concepts.** We show examples of low-entropy concepts that fire solely for DinoV2. These concepts fire for perspective cues related to view invariance such as converging perspective lines (concept 3715 and 4189) and angled scenes (concept 2562 and 3003).

while 'brown grass' activates on grass in our heatmaps, some models' activation maximizations include birds, suggesting animals also influence the concept's activation.

### 4.6. Discovering Unique Concepts with USAEs

Our universal training objective provides us the opportunity to explore concepts that may arise independently in one model, but not in others. Using metrics for universality, Eqs. 13 and 12, we can search for concepts that fire with a *low entropy*, thereby isolating firing distributions whose probability mass is allocated to a single model. We explore this direction by isolating unique concepts for DinoV2 and SigLIP, both of which have been studied for their unique generalization capabilities to different downstream tasks (Amir et al., 2022; Zhai et al., 2023).

#### 4.6.1. UNIQUE DINOV2 CONCEPTS

DinoV2's unique concepts are presented in Figures 9 and 10. Interestingly, we find concepts that solely fire for DinoV2 related to *perspective* and *depth* cues. These features follow surfaces and edges to vanishing points as in concept 3715 and 4189, demonstrating features for converging perspective lines. Further, in Figure 10 we find features for object groupings placed in the scene at varying depths in concept 4756, and background depth cues related to uphill slanted surfaces in concept 1710. We also find features that suggest a representation of view invariance, such as concepts related to the angle or tilt of an image (Fig. 9) for both left (concept 3003) and right views (concept 2562). Lastly, we observe unique geometric features in Fig. 11 that suggest some low-level 3D understanding, such as concept 4191 that fires for the top face of rectangular prisms, concept 3448 for brim lines that belong to dome shaped objects, as well as concept 1530 for corners of objects resembling rectangular prisms.

View invariance, depth cues, and low-level geometric concepts are all features we expect to observe unique to Di-

noV2's training regime and architecture (Oquab et al., 2023). Specifically, self-distillation across different views and crops at the image level emphasizes geometric consistency across viewpoints. This, in combination with the masked image modelling iBOT objective (Zhou et al., 2021) that learns to predict masked tokens in a student-teacher distillation framework, would explain the sensitivity of DinoV2 to perspective and geometric properties, as well as view-invariant features. We further explore unique concepts in SigLIP using this same approach in A.3 finding concepts that fire for both visual and textual elements of the same concept.

## 5. Conclusion

In this work, we introduced *Universal Sparse Autoencoders* (USAEs), a framework for learning a unified concept space that faithfully reconstructs and interprets activations from multiple deep vision models at once. Our experiments revealed several important findings: (i) qualitatively, we discover diverse concepts, from low-level primitives like colors, shapes and textures, to compositional, semantic, and abstract concepts like groupings, object parts, and faces, (ii) many concepts turn out to be both *universal* (firing consistently across different architectures and training objectives) and *highly important* (responsible for a large proportion of each model's reconstruction), (iii) certain models, such as DinoV2, encode unique features even as they share much of their conceptual basis with others, and (iv) while universal training recovers a significant fraction of the concepts learned by independent single-model SAEs, it also uncovers new shared representations that do not appear to emerge in model-specific training. Finally, we demonstrated a novel application of USAEs—*coordinated activation maximization*—that enables simultaneous visualization of a universal concept across multiple networks. Altogether, our USAE framework offers a practical and powerful tool for multi-model interpretability, shedding light on the commonalities and distinctions that arise when different architectures, tasks, and datasets converge on shared high-level abstractions.

## Impact Statement

This work advances interpretability for machine learning systems. More specifically, understanding *shared* representations across deep neural networks (DNNs) is essential for scalable interpretability, enabling more effective risk mitigation, robust model design, and compliance with evolving regulations. By moving beyond single-model analysis, we aim to help establish a unified framework for interpreting diverse architectures, fostering transparency and accountability in AI deployment.

## Acknowledgements

This work was completed with support from the Vector Institute, and was funded in part by the Canada First Research Excellence Fund (CFREF) for the Vision: Science to Applications (VISTA) program (K.G.D, H.T), the NSERC Discovery Grant program (K.G.D), the NSERC Canada Graduate Scholarship Doctoral program (M.K), the Ontario Graduate Scholarship (H.T), and a gift from the Chan Zuckerberg Initiative Foundation to establish the Kempner Institute for the Study of Natural and Artificial Intelligence at Harvard University (T.F).

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

# A. Appendix

## A.1. SAE Training Implementation details

We modify the TopK Sparse Autoencoder (SAE) (Gao et al., 2024) by replacing the $\ell_2$ loss with an $\ell_1$ loss, as we find that this adjustment improves both training dynamics and the interpretability of the learned concepts. The encoder consists of a single linear layer followed by batch normalization (Ioffe & Szegedy, 2015) and a ReLU activation function, while the decoder is a simple dictionary matrix.

For all experiments, we use a dictionary of size $8 \times 768 = 6144$ which is an expansion factor of $8$ multiplied by the largest feature dimension in any of the three models, $768$. All SAE encoder-decoder pairs have independent Adam optimizers (Kingma & Ba, 2015), each with an initial learning rate of $3e-4$, which decays to $1e-6$ following a cosine schedule with linear warmup. To account for variations in activation scales caused by architectural differences, we standardize each model's activations using 1000 random samples from the training set. Specifically, we compute the mean and standard deviation of activations for each model and apply standardization, thereby preserving the relative relationship between activation magnitudes and directions while mitigating scale differences.

Since SigLIP does not incorporate a class token, we remove class tokens from DinoV2 and ViT to ensure consistency across models. Additionally, we interpolate the DinoV2 token count to match a patch size of $16 \times 16$ pixels, aligning it with SigLIP and ViT. We train all USAEs on a single NVIDIA RTX 6000 GPU, with training completing in approximately three days.

## A.2. Unique DinoV2 Concepts

DinoV2's unique concepts are presented in Figures 10 and 11. Interestingly, we find concepts that solely fire for DinoV2 related to *depth*, *perspective*, and *geometric* cues.

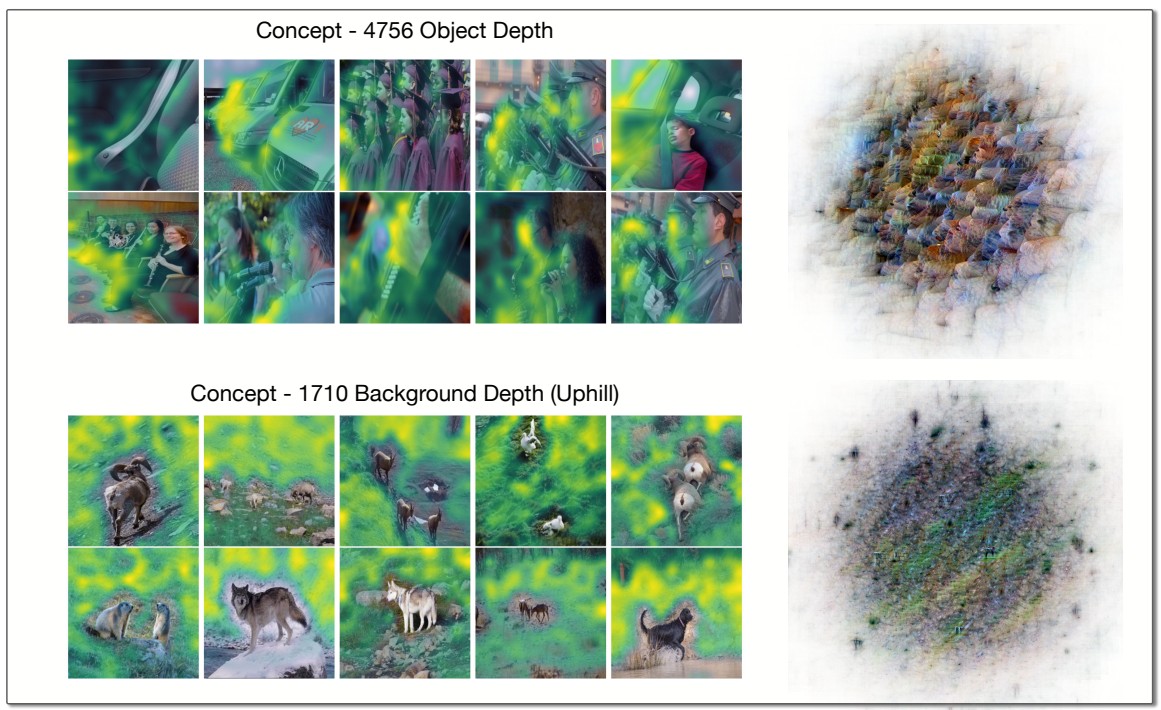

*Figure 10.* **Qualitative results of low-entropy concepts that fire for DinoV2.** We discover features related to depth cues for foreground objects as well as background in concept 4756 (above) and 1710 (below).

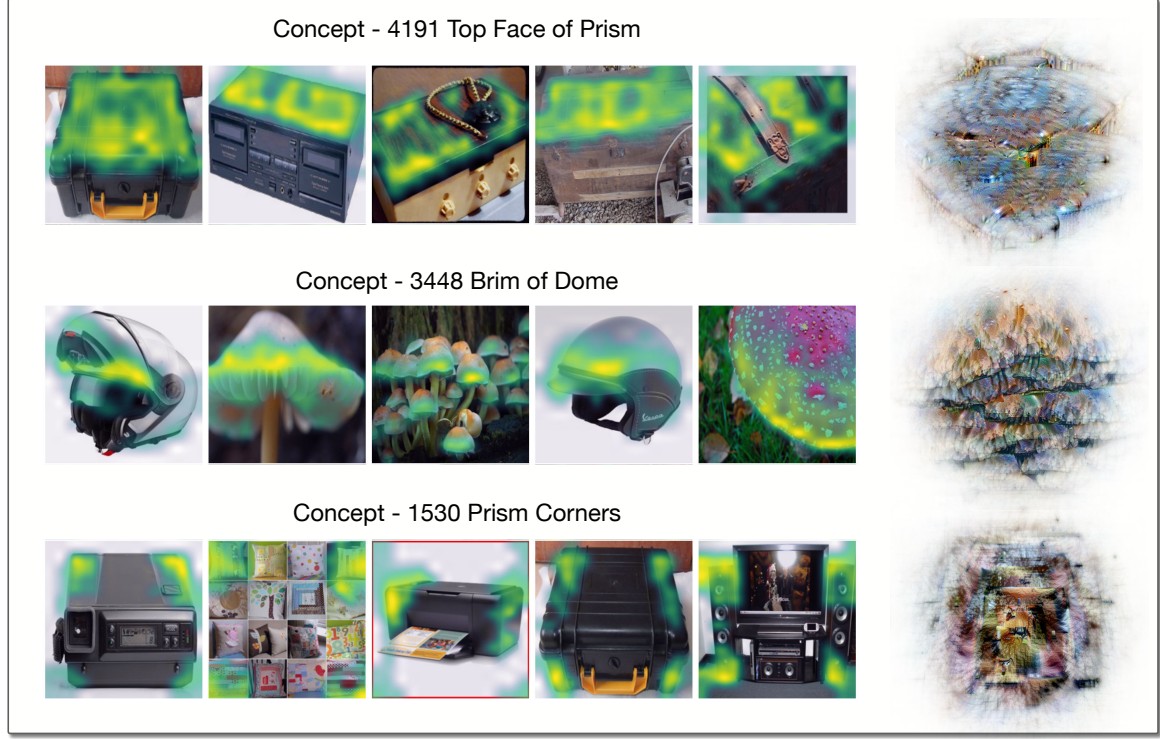

*Figure 11.* **Qualitative results for low-entropy concepts that fire for DinoV2.** We discover DinoV2 independent features that are not universal suggesting 3D understanding like corners (concepts 1530), top face of rectangular prism (concept 4191), and brim of dome (concept 3448).

### A.3. Unique SigLIP Concepts

Similar to DinoV2, we isolate concepts with low firing-entropy where probability mass is concentrated for SigLIP. Example concepts are presented in Fig. 12. We observe concepts that fire for both visual and textual elements of the same concept. Concept 5718 fires for both the shape of a star, as well as regions of images with the word or even just a subset of letters on a bottlecap and sign at different scales. Additionally, concept 2898 fires broadly for various musical instruments, as well as music notes, while concept 923 fires for the letter 'C'. For each of these concepts, the coordinated activation maximization visualization has both the physical semantic representation of the concept, as well as printed text. The presence of image and textual elements are expected given SigLIP is trained as a vision-language model with a contrastive learning objective, where the aim is to align image and text latent representations from separate image and language encoders. While we do not train on any activations directly from the language model, we still observe textual concepts in our image-space visualizations.

### A.4. Out-of-Distribution Generalization

In order to assess the out-of-distribution capabilities of our approach, we use DTD (Cimpoi et al., 2014) and CelebA (Liu et al., 2015) as the validation dataset for our ImageNet trained USAEs and show strong evidence of generalization outside of the training distribution as seen in Table 1. We find consistent activation reconstruction accuracy (measured by MSE and R2), consistent trends in co-firing metrics in Fig. 13 and visualize some of the most important concepts for these new datasets, along with their associated highest activating images, from ImageNet in Fig. 14 and 15. Despite differences in domain and semantics, USAEs trained on ImageNet exhibited robust generalization to both DTD and CelebA. Importantly, many of the concepts identified in these datasets also aligned with high-activation concepts from ImageNet, suggesting that the USAE dictionary captures generalizable structure beyond its training data.

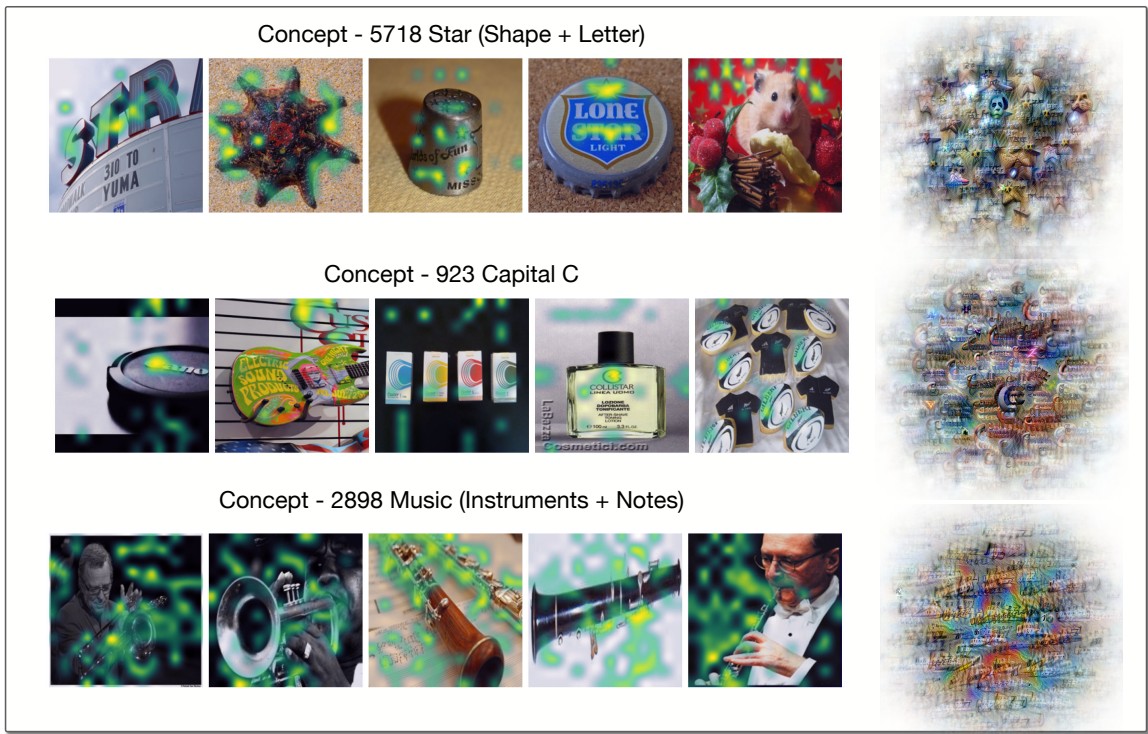

*Figure 12.* **Qualitative results of low-entropy SigLIP concepts.** We consistently find concepts that fire for abstract concepts in image space such as images or text of 'star' (concept 923), letters (concept 5718), and music notes (concept 2958).

| Model | Mean Squared Error (MSE) ↓ | | | Coefficient of Determination (R²) ↑ | | |
|---|---|---|---|---|---|---|
| | ImageNet | DTD | CelebA | ImageNet | DTD | CelebA |
| SigLIP | 0.39 | 0.38 | 0.48 | 0.61 | 0.54 | 0.52 |
| DinoV2 | **0.19** | **0.26** | **0.22** | **0.77** | **0.69** | **0.75** |
| ViT | 0.41 | 0.56 | 0.69 | 0.59 | 0.46 | 0.45 |

*Table 1.* Performance comparison of vision models across different datasets. Lower MSE and higher R² indicate better performance.

## A.5. Additional Results

### A.5.1. ADDITIONAL QUANTITATIVE RESULTS

Figure 16 presents concept consistency distributions across models for the top 1,000 co-firing concepts. We find an overall increase in consistency between individual and universal concepts: the most universal (highest co-firing) concepts are more likely to be found in each model's respective independently trained SAE. Within this thresholded range, we find DinoV2 to exhibit the highest similarity between individual and universal concepts with an average cosine similarity of 0.65, followed by ViT at 0.52 and SigLIP at 0.40. DinoV2 concepts seem to be better represented in the universal space, suggesting that some models may have more universal concepts than others.

### A.5.2. ADDITIONAL QUALITATIVE RESULTS

We provide additional universal concept visualizations for the top activating images for that concept across each model. Specifically, we showcase low-level concepts in Fig. 17 related to texture like shell and wood for concepts 1716 and 2533, respectively, as well as tiling for concept 5563. We also showcase high-level concepts in Fig. 18 related to environments like auditoriums in concept 4691, object interactions like ground contact in concept 5346, as well as facial features like snouts in concept 3479.

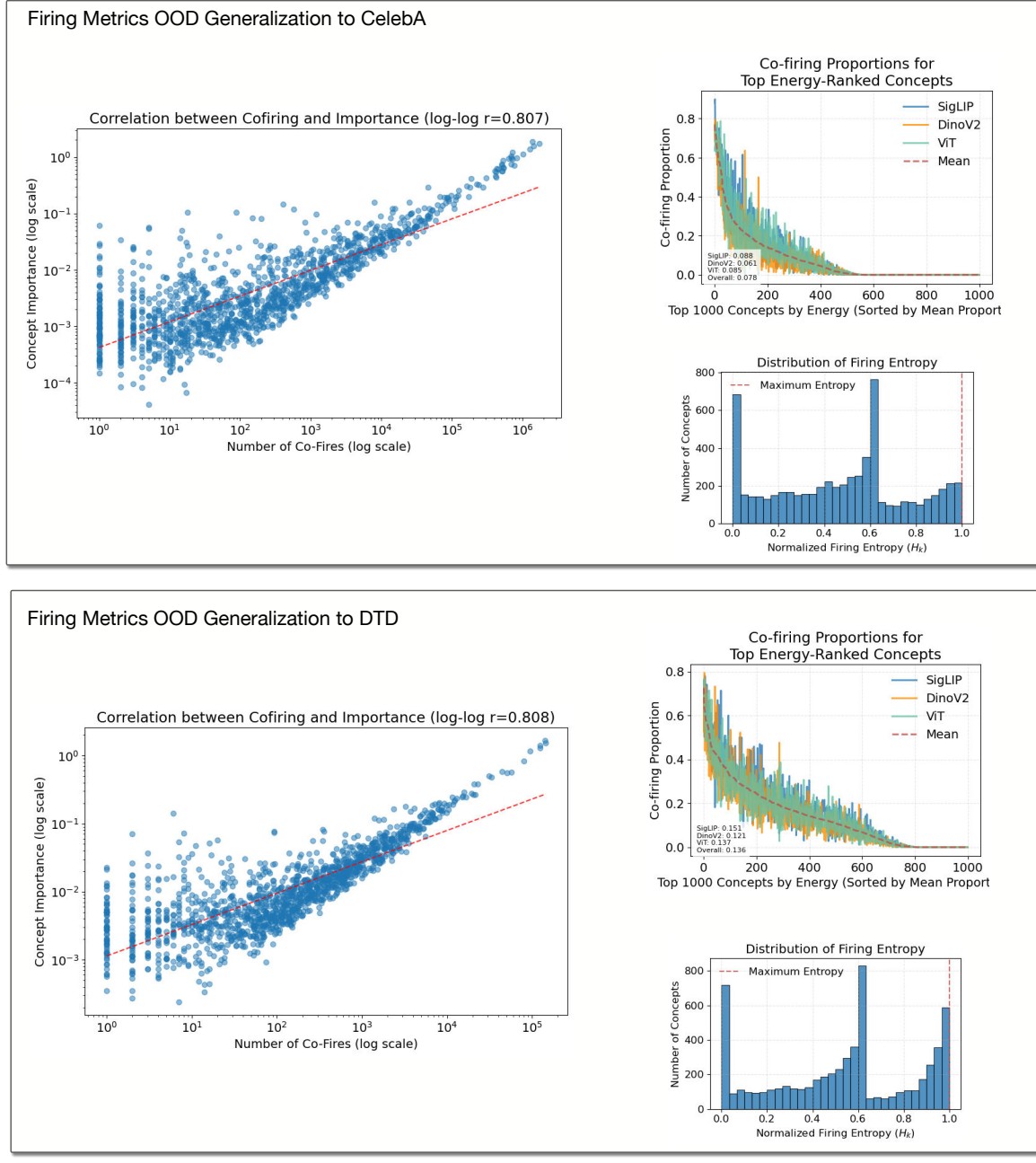

*Figure 13.* **Zero-shot generalization quantitative results of universal concepts on out-of-distribution datasets. Top:** When applying our ImageNet trained USAE to the validation set of CelebA we find consistent trends across each of our universality metrics. We find a clear correlation between co-firing and concept importance. The distribution over firing entropy also indicates concepts that fire uniquely for a single model, two of three models, and universal concepts that fire for all three. **Bottom:** When applying our ImageNet trained USAE to the validation set of DTD we find consistent trends across each of our universality metrics. We find a clear correlation between co-firing and concept importance. The distribution over firing entropy is tri-modal, indicating concepts that fire uniquely for a single model, two of three models, and universal concepts that fire for all three.

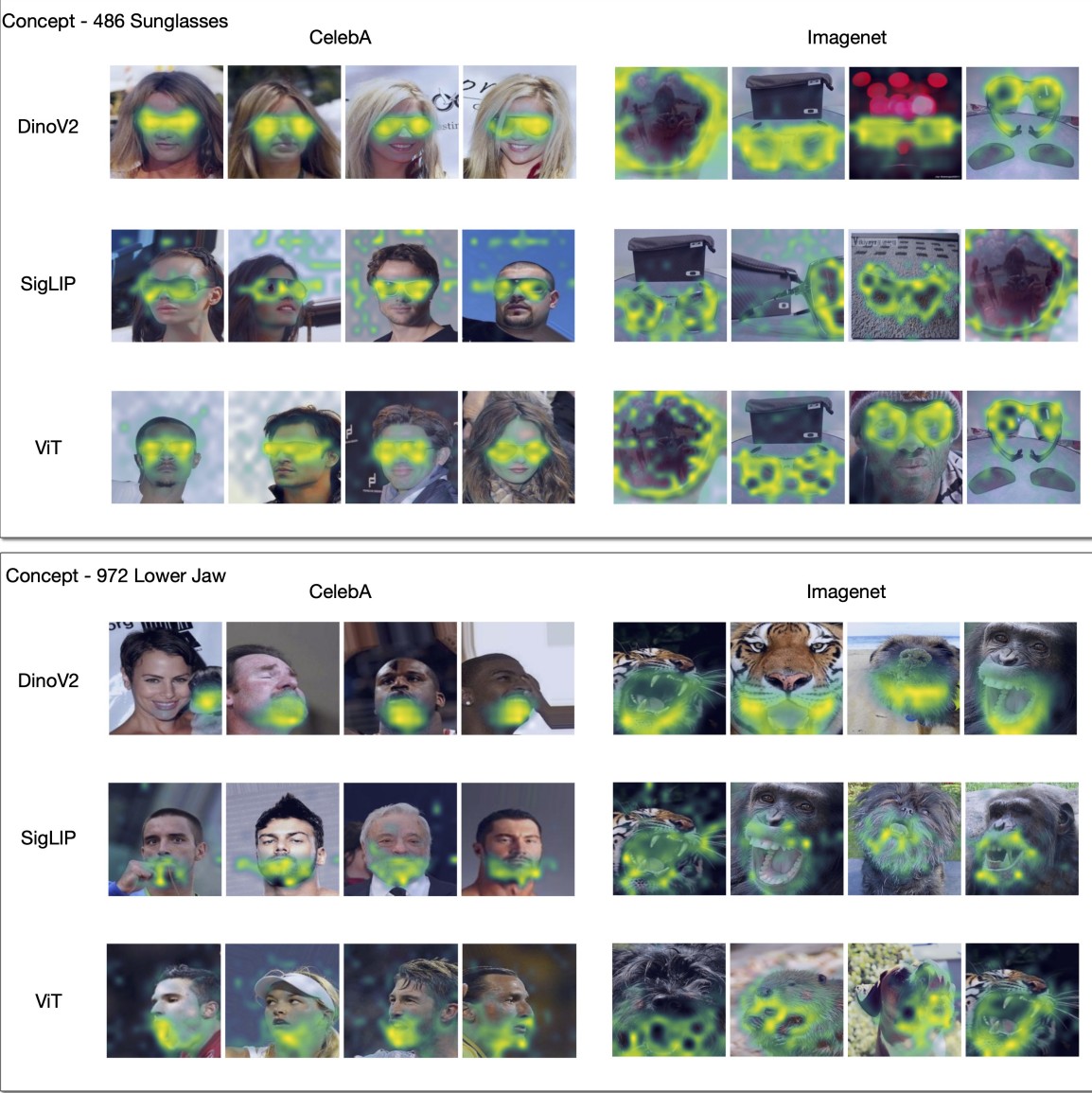

*Figure 14.* **Qualitative examples of zero-shot generalization of ImageNet trained USAE on CelebA. Top:** We depict high-level visual concept dimension related to sunglasses and the highest activating images for the validation sets of both ImageNet and CelebA. **Bottom:** We depict the lower-jaw concept's highest activating images for the validation set of ImageNet and CelebA. This jaw concept generalizes beyond animal jaw to include human jaws as seen from our CelebA heatmaps.

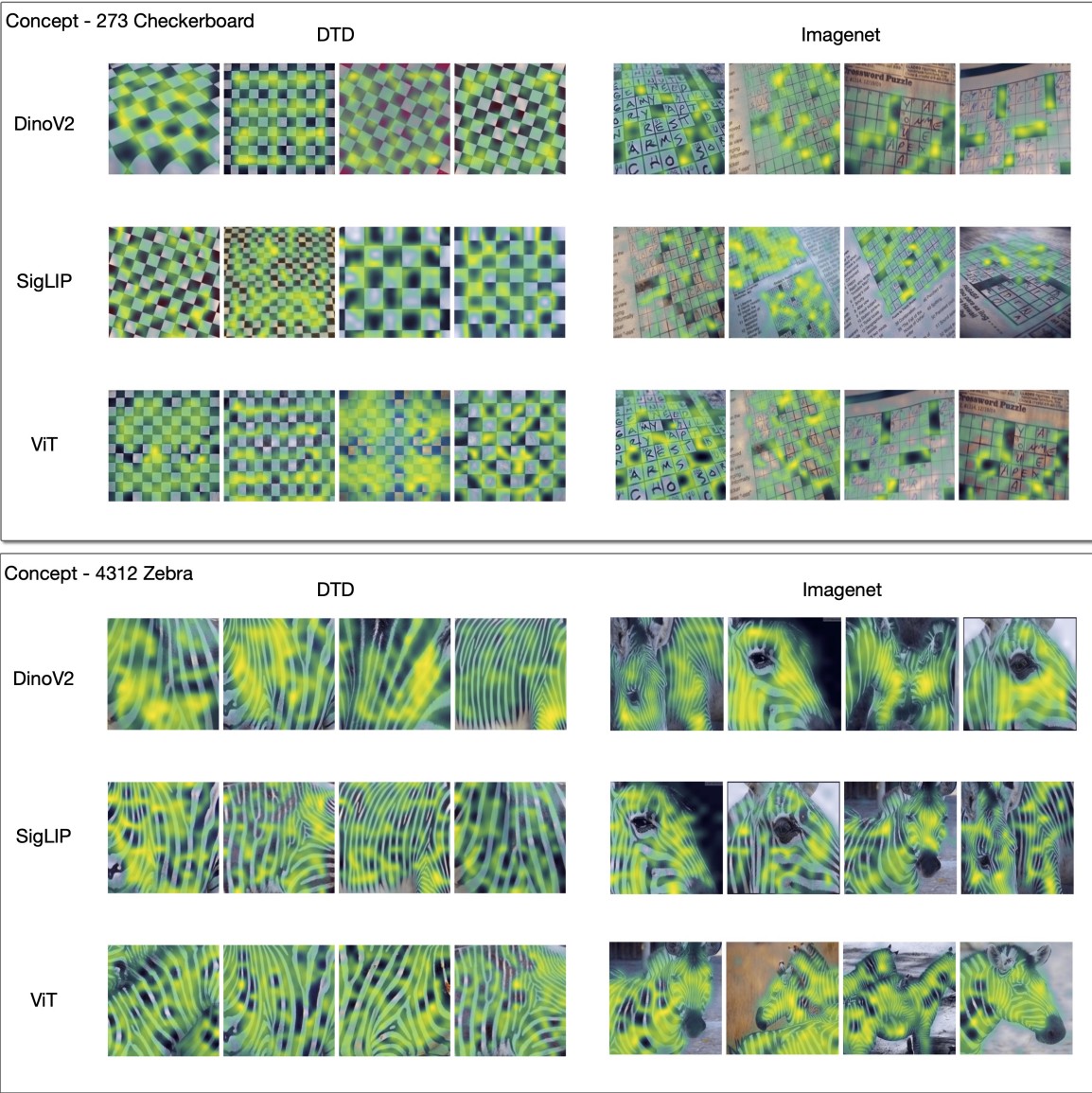

*Figure 15.* **Qualitative examples of zero-shot generalization of ImageNet trained USAE on DTD. Top:** We depict a checkerboard concept's highest activating images for the validation set of ImageNet and DTD. This checkerboard concept generalizes from low-level textures in DTD like tiles to sudoku and crossword puzzles in ImageNet. **Bottom:** We depict a concept for zebra stripes and its highest activating images for the validation set of ImageNet and DTD. This stripe concept generalizes across scales for images zoomed in on the animal in DTD to across a whole zebra in ImageNet.

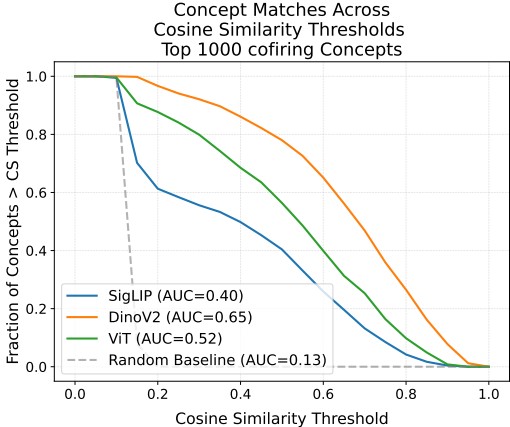

*Figure 16.* **Top 1000 co-firing concept consistency between independent SAEs and Universal SAEs.** Our universal training objective discovers universal concepts that have overlap (i.e., cosine similarity) with those discovered with independent training. In descending order, Universal SAEs have highest overlap with independently trained DinoV2, ViT, and SigLIP. The smaller overlap observed with SigLIP suggests the aligned image-language embedding space produces unique concepts that are more distinct from those in DinoV2 and ViT.

### A.6. Limitations

Our universal concept discovery objective successfully discovers fundamental visual concepts encoded between vision models trained under distinct objectives and architectures, and allows us to explore features that fire distinctly for a particular model of interest under our regime. However, we note some limitations that we aim to address in future work. We notice some sensitivity to hyperparameters when increasing the number of models involved in universal training, and use hyperparameter sweeps to find an optimal configuration. We also constrain our problem to discovering features at the last layer of each vision model. We choose to do so as a tractable first step in this novel paradigm of *learning* to discover universal features. We leave an exploration of universal features across different layer depths for future work. Lastly, we do find qualitatively that a small percentage of concepts are uninterpretable. They may be still stored in superposition (Elhage et al., 2022) or they could be useful for the model but simply difficult for humans to make sense of. This is a phenomena that independently trained SAEs suffer from as well. Many of the limitations of our approach are tightly coupled to the limitations of training independent SAEs, an active area of research.

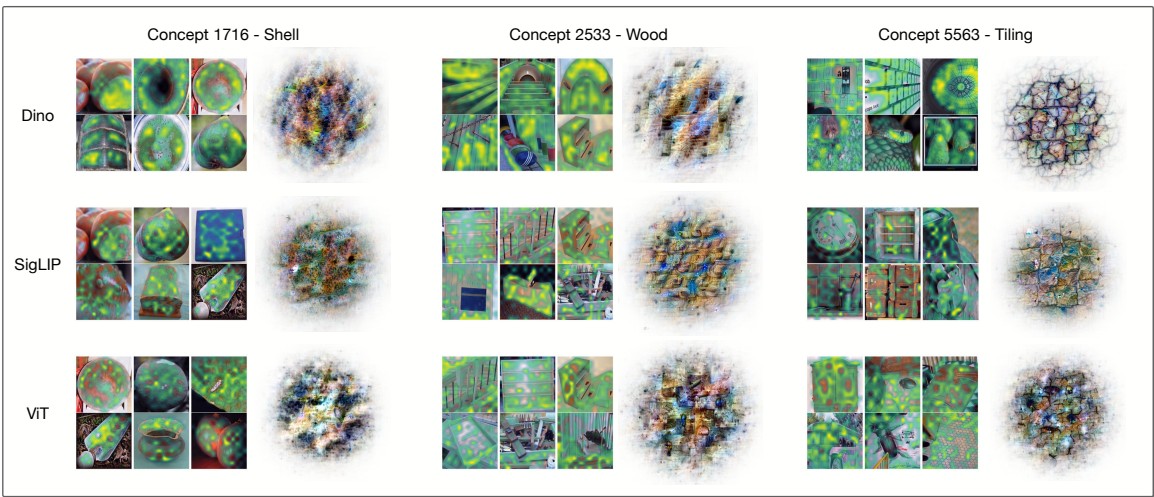

*Figure 17.* **Qualitative results of universal concepts.** We depict low-level visual features related to textures, such as shells (concept 1716), wood (concept 2533), and tiling (concept 5563).

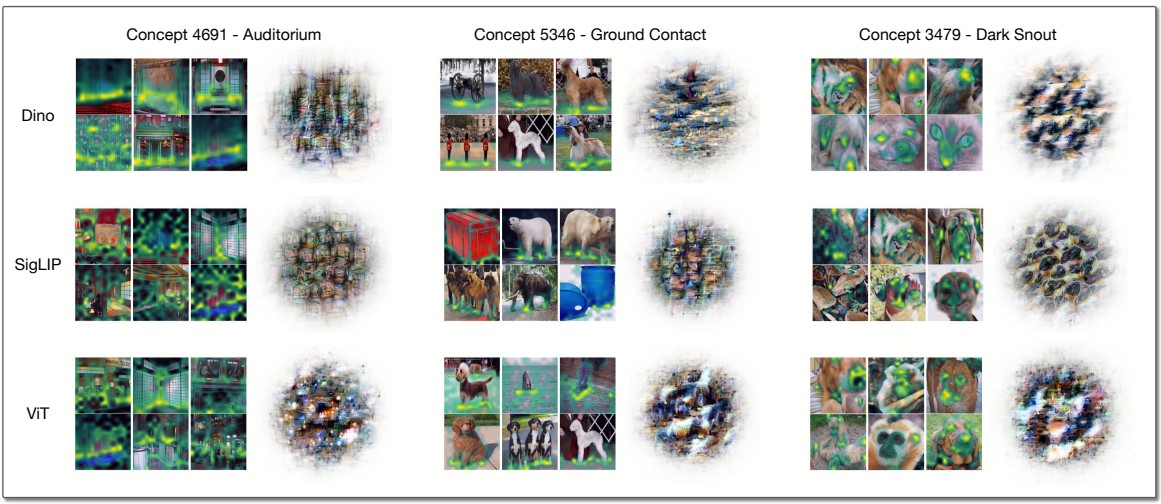

*Figure 18.* **Qualitative results of universal concepts.** We depict high-level visual features related to environments, such as auditoriums (concept 4691), ground contact (concept 5346), and animal snouts (concept 3479).

