# OpenReview forum: "Universal Sparse Autoencoders: Interpretable Cross-Model Concept Alignment"
_ICML.cc/2025/Conference — ICML 2025 poster_

### Official Review · Reviewer_SLTN · 2025-03-12

**Overall Recommendation:** 4

**Summary:**

The submission is based on learning a single concept space that is shared between SAEs trained on multiple vision models.  The aim is to learn a universal set of concepts which can be used to translate between different models and highlight differences in how models represent visual information.

The experiments center on a universal concept space trained on a set of three models (SigLIP, ViT, and DinoV2).  Various characterizations are performed: the dominant concepts are visualized via activation heatmaps; the reconstruction quality across and within each model; concept firing distributions; comparison to singular SAEs; and universal concept activation maximization.

**Claims And Evidence:**

The paper convincingly supports the claim that a shared concept space was learned, and I agree that the analysis is of a depth that novel insights are offered about how these different models represent visual information.  The claim of novelty around “coordinated activation maximization” appears overblown -- it’s activation maximization on the learned space, repeatedly applied.

**Essential References Not Discussed:**

No, the references appear quite thorough.

**Experimental Designs Or Analyses:**

The paper provides little information about the experimentation that culminated in the final setup, which weakens the presentation of the method.

Can any evidence at all be presented for L1 improving interpretability over L2 in the topK SAE implementation?

How many epochs were needed for convergence?

The interpolation of the size 14 patches used in dinoV2 to the size 16 patches for the other two models is reasonable, though it would be great to have more of a discussion and demonstration of the effect of this interpolation vs any other form.  Was the interpolation bilinear?

What were the specific versions of the models used, given that `timm` has multiple for each?

In the top-K SAE implementation, what is k and how was it determined?

**Methods And Evaluation Criteria:**

Largely, yes.  There are a few issues which would strengthen the paper if addressed:

- Why introduce a firing threshold $\tau$ when using a top-k SAE?  In other words, we know exactly what concepts fire for each input -- it’s the top k.  By introducing a degree of arbitrariness, and one that is decoupled from the functioning of the SAE, subsequent processing is called into question.

- The firing entropy does not actually use firing patterns across the models; it merely compares counts.  A concept could fire completely disjointly across the models (say, 20 times for each model on a total of 60 inputs) and the paper’s firing entropy metric would be maximized.  Thus the evaluation is problematic, and might be improved by directly assessing the firing distribution per concept.  This would be easy -- each firing pattern is a three bit vector, and the authors could calculate something like the total correlation of the joint distribution from the 8 probability values.

**Other Comments Or Suggestions:**

Could it provide a useful frame of reference to train a USAE with a known mismatch, such as with an early and a late layer of the same model?  The sense in which concepts are shared or not might be clearer, in order to build a better understanding of the method before application to varied vision models.  I understand this is most likely too much for the rebuttal period, so commenting how the authors see this experiment playing out would be fine.

**Other Strengths And Weaknesses:**

The paper is nicely written and the analyses are well-motivated, even if there is some room for improvement.

**Questions For Authors:**

- Regarding reconstruction $R^2$ values, the paper states “positive off-diagonal $R^2$ scores indicate successful cross-model reconstruction...” (L313).  Merely having a positive $R^2$ is an extremely low bar for success.  The off-diagonal values are around 0.3-0.4 -- which still does not seem particularly successful.  Can the authors provide more grounding for why this is good enough, and not higher?

**Relation To Broader Scientific Literature:**

The paper leverages SAEs to connect multiple vision models into a shared space -- a nice idea given that SAEs are increasingly used to gain interpretability in individual models.

**Theoretical Claims:**

I saw no theoretical claims.

---

> ### Author Rebuttal · Authors · 2025-04-01
>
> We thank the reviewer for their thorough analysis of our work, and thank them for finding our paper is nicely written with well-motivated analyses. We aim to answer the questions raised below. For plots and figures, please refer to: https://sparkling-queijadas-998747.netlify.app/
>
> ### **Implementation Details**
> *“Can any evidence at all be presented for L1 improving interpretability over L2?”*
> In our early experiments, we found that training USAEs with an L1 reconstruction loss tended to yield more interpretable concepts than L2—particularly in the sparsity and visual clarity of activations. While this remains a qualitative observation, it aligns with prior findings in cross-model interpretability settings [Lindsey et al., 2024]. We emphasize that USAEs are not tied to a specific loss; we simply chose L1 based on these initial results and precedent.
>
> *“How many epochs were needed for convergence?”*
> All USAEs were trained for 30 epochs, after which reconstruction metrics plateaued and we observed diminishing returns. This setting was consistent across all experiments.
>
> *“Was the interpolation bilinear?”*
> We appreciate the suggestion and agree that the token interpolation choice is an interesting direction. We used bilinear interpolation in our experiments. Its main effect appears as smoothing in the border regions of attribution maps. However, we haven't observed meaningful differences in the types of concepts discovered. We plan to explore alternative schemes (e.g., bicubic, with and without anti-aliasing) in future work.
>
> *“What were the specific versions of the model used?”*
> SigLIP: 'vit_base_patch16_siglip_224'
> DinoV2: ‘facebookresearch/dinov2_vits14’
> ViT: ‘'vit_base_patch16_224'
>
> *“In the top-K SAE implementation, what is k and how was it determined?”*
> We set K = 32 for all experiments, following prior work on sparse autoencoders [Gao et al., 2024]. While this choice worked well empirically, we acknowledge that concept interpretability and stability may vary with K, and we leave a deeper hyperparameter sweep to future work.
>
> ### **Firing Threshold $\tau$**
> We agree that one could just use the K largest dictionary entries; however, our use of a threshold was to reduce noise in the case for which some of the top K firing dimensions are very close to 0. For example, some concepts may be captured successfully with only a subset of K dimensions, which could be identified with a small threshold to ensure the magnitude of the activation is sufficient.
>
> ### **Firing Entropy vs Co-Fire Proportion**
> We agree with the reviewer that modifying the firing-entropy (FE) metric to measure the distribution of fires per concept between models may yield additional useful insights. The FE metric focuses on firing counts (irrespective of co-fires) to initially probe whether concepts were particularly biased to any of the models. We choose to measure this to avoid degenerate solutions of reserving subsets of concept dimensions for specific models. Once we determined that many concepts fired with equal probability for each model, we used co-firing proportion (CFP) as a more granular token-level analysis to see if these concepts fired for the same tokens.
>
>
> ### **Training on Different Layers for Frame of Reference**
> We appreciate the suggestion to train our USAE on the first and last layers of a single network as a baseline. We expect partial concept overlap between layers, as shown in Fig. 4 where low-level features (e.g., color blue, concept 4235) emerge from last-layer training. Our DTD experiments (**Generalization OOD** 51dn) further demonstrate detection of low-level concepts when applying ImageNet-trained USAE to texture data. Given neural networks' hierarchical feature learning, finding low-level features in the last layer suggests our method may identify "layer-redundant" features encoded across multiple network depths. This makes cross-layer training an imperfect mismatch but still a valuable experiment.
>
> ### **R2 off-diagonal scores**
> In the confusion matrix (Fig. 5), the maximum possible off-diagonal R2 score is 1, which would imply that activations from one model (e.g., SigLIP) can be perfectly reconstructed by encoding activations from a different model (e.g., DinoV2 or ViT). While the upper bound for such cross-model reconstruction is unclear, our positive off-diagonal R2 scores already provide strong evidence of shared structure across models. These results suggest that USAEs capture meaningful, transferable representations even across architectures and training paradigms.
>
> We believe that further optimizing the USAE design—including architecture (increasing the depth of the encoder, Matching Pursuit encoder…), loss functions (HSIC, Cosine, …), training objectives (iBot style…), and hyperparameters—could improve cross-model reconstruction and raise this empirical bound. We leave a more detailed exploration of these design factors to future work.

---

### Official Review · Reviewer_42Tj · 2025-03-14

**Overall Recommendation:** 3

**Summary:**

The authors introduce USAEs, a framework that jointly learns a universal concept space for the internal activations of multiple vision models. By optimizing a shared objective, they show that USAEs semantially coherent universal concepts at different levels across vision models. Their results showcase the strong correlation between concept universality and importance, and also identify unique features that are learned by individual models. A unique application of USAEs, coordinated activation maximization, is also presented to achieve simultaneous visualization of universal concepts across models.

**Claims And Evidence:**

Yes

**Essential References Not Discussed:**

No

**Experimental Designs Or Analyses:**

Yes

**Methods And Evaluation Criteria:**

Yes

**Other Comments Or Suggestions:**

1. While $x\in \mathcal{X}$ is written in Equation (8), I don't see where $x$ is used in the equation. Is it the same as $x^{(i)}$?
2. Where are the results for the discussion in Line 317?

**Other Strengths And Weaknesses:**

Strengths:
1. The introduction of USAEs is novel, exploring a new direction of learning universal SAE concepts across models
2. Extensive experiments are done to demonstrate the capture of universal concepts using USAEs.
3. The presentation is clear and easy to follow.

Weaknesses:
1. Some implementation details are not clear (e.g., what is the threshold used in Figure 5; how are the concept meanings summarized).
2. The comparison between USAEs and independent SAEs is not sufficient (e.g., do they find more universal concepts than just comparing independent SAEs; will USAEs show more consistent visual results?).
3. Some experiments need more discussions (e.g., what is the connection between FE and CFP?)

**Questions For Authors:**

1. Is Energy(k) (Equation (9)) computed for just one model? Or is it computing a universal score for the concept k across models?
2. A baseline would be trying to identify shared concepts in independent SAEs. Do USAEs find more universal concepts than just comparing independent SAEs? Also, there is a lack of visual comparison between USAEs and SAEs.
3. Will USAEs learn more redundant concepts than SAEs?
4. Although the authors claim Equation (6) will "strike a practical balance between training speed and memory usage", there is no empirical evidence supporting this.
5. It is not clear to me how the concept meanings shown in this paper were generated. Were they manually summarized by humans?
6. How is the threshold determined for results in Figure 5.
7. More discussion/analysis should be made on the connection between FE and CFP. Will the high value of one metric always lead to the high value of the other?
8. Why do the results in line 357 indicate that SigLIP and ViT share more concepts?

**Relation To Broader Scientific Literature:**

While previous papers focus on learning the concepts in individual models with SAEs, this paper propose a new USAE for learning universal concepts across models.

**Theoretical Claims:**

N/A

---

> ### Author Rebuttal · Authors · 2025-04-01
>
> We thank the reviewer for their thoughtful analysis of our work, and appreciate the positive compliments regarding the presentation quality, novelty, and extensive experimental results. We answer the questions raised below in text form. For plots and figures, please refer to our temporary anonymous website hosted on the Netlify platform: https://sparkling-queijadas-998747.netlify.app/
>
>
> ### **USAE vs SAE Features**
> See our response **USAE vs SAE features** to (51dn) for further discussion.
>
> ### **Firing Entropy vs Co-Firing Proportion**
> See our response **Firing Entropy vs Co-Fire Proportion** to (SLTN)
>
>
> ### **Clarifying Questions**
>
> 1. *Is Energy(k) (Equation (9)) computed for just one model? Or is it computing a universal score for the concept k across models?*
>
>     Energy is first computed individually across each model and we take the average to rank concepts.
>
> 2. *A baseline would be trying to identify shared concepts in independent SAEs. Do USAEs find more universal concepts than just comparing independent SAEs? Also, there is a lack of visual comparison between USAEs and SAEs.*
>
>     Please see **USAE vs SAE features** in our response to (51dn)
>
> 3. *Will USAEs learn more redundant concepts than SAEs?*
>
>     Assuming redundant concepts refers to when multiple concepts that look visually similar and appear repeatedly, we do not observe many of them in the learned dictionary. Designing a formal way of quantifying their frequency and performing a deeper analysis of their frequency is an interesting direction to explore for future work.
>
> 4. *Although the authors claim Equation (6) will "strike a practical balance between training speed and memory usage", there is no empirical evidence supporting this.*
>
>     The alternative approach for mining universal concepts between individual SAE’s is restricted to being pairwise [Lan et al. 2024]; scaling this approach greatly increases its computational complexity. Our approach, being learning-based, maintains the same complexity at the cost of training time as we increase the number of models used.
>
> 5. *It is not clear to me how the concept meanings shown in this paper were generated. Were they manually summarized by humans?*
>
>     Concept meanings were determined by qualitative inspection, as is common practice in previous works  [Ghorbani et al. 2019, Fel et al. 2023c, Kowal et al. 2024b]. However, we believe the rise of more capable vision+language models (e.g., Vision-LLMs) could aid in automated summarization of the results.
>
> 6. *How is the threshold determined for results in Figure 5?*
>
>     Assuming the threshold being referred to is Figure 5. (C): We observed a clear phase transition beyond 1000 concepts (e.g., r=0.63 vs. r=0.89). Thus, we aimed to analyze properties of these highest co-firing concepts and set the threshold to >1000 highest co-firing concepts.
>
> 7. *More discussion/analysis should be made on the connection between FE and CFP. Will the high value of one metric always lead to the high value of the other?*
>
>     See (SLTN) Firing Entropy vs Co-Fire Proportion
>
> 8. *Why do the results in line 357 indicate that SigLIP and ViT share more concepts?*
>
>     Phrased differently, the results in L357 indicate that SigLIP and ViT share a higher fraction of total concepts which co-fire across all three models, this is likely due to DinoV2 possessing a higher fraction of unique concepts due to its training objective promoting 3D scene understanding, the concepts of which we observe in appendix Fig.10 and 11 encoded as low-entropy concepts.
>
> ### **Implementation Details**
> See response to (SLTN) **implementation details**.

---

### Official Review · Reviewer_51dn · 2025-03-14

**Overall Recommendation:** 3

**Summary:**

This work introduces Universal Sparse Autoencoders (USAEs), a method to discover concepts shared across different deep learning models. The authors focus on the study of USAEs for the last-layer representations of three popular vision models, showing that the methodology enables the construction of an interpretable feature space shared across different architectures. The authors present a coordinated activation maximization procedure to investigate concept alignment. The paper supports that the proposed procedure produces meaningful universal features through qualitative and quantitative analyses.

**Claims And Evidence:**

The claims in the paper are well supported by experimental evidence.
Both qualitative (Fig 1) and quantitative (Fig 3 and 4) results support the evidence that USAEs learn meaningful features shared among the three models.
The claim that USAEs learn features at different levels of granularity is supported by clear qualitative evidence.

The work would benefit from further evidence that the method allows the construction of truly universal features. Considering that all three models have been trained on Imagenet, the claim would be further strengthened by showing that this is still true for images not in the training set.

**Essential References Not Discussed:**

To the reviewer's knowledge, there are not major papers that have been overlooked by the authors.

**Experimental Designs Or Analyses:**

The cross-model reconstruction, co-firing rate, and energy-based importance metrics are appropriate for measuring concept alignment. Qualitative visualizations effectively illustrate shared concepts.

**Methods And Evaluation Criteria:**

The method proposed by the authors is suited for the purpose of constructing interpretable features that are shared among different vision models.

The evaluation criteria are generally well constructed but can be strengthened in some parts. For example, in Section 4.4, the authors could better characterize what type of features the USAE method allows to learn with respect to SAE features, which are shared among the different models. The analysis in this section would benefit from a more clear description.

**Other Comments Or Suggestions:**

I have no further comments or suggestions.

**Other Strengths And Weaknesses:**

The authors propose an original, well-structured way to analyze shared representations across multiple architectures, which leads to interesting observations concerning model alignment.
The Coordinated Activation Maximization method proposed in the work is a useful strategy for comparing concept representations across different models.
The presentation of the results and the definition of the necessary mathematical concepts are explained with a good level of clarity.
The work would benefit from further evidence that the feature identified by the USAE methodology generalize to other dataset and are not restricted to the three models considered in the work.
A more thorough discussion of the advantages of the proposed strategy in terms of scalability could benefit the work, specially in the light of the fact that the authors do not outline (at least in terms of a proof of concept) automatic strategies to interpret USAEs coordinates beyond visual inspection.

**Questions For Authors:**

I have no further questions for the authors.

**Relation To Broader Scientific Literature:**

The authors contextualize well their work within existing literature.

**Theoretical Claims:**

NA

---

> ### Author Rebuttal · Authors · 2025-04-01
>
> We thank reviewer 51dn for their thorough review of our work. We appreciate that they found the paper original and well-structured, and that they found the proposed Coordinated Activation Maximization application a promising tool for concept visualization. For plots and figures, please refer to our temporary anonymous website hosted on the Netlify platform: https://sparkling-queijadas-998747.netlify.app/
>
> ### **Dataset Generalization (OOD)**
> We agree that testing for potential dataset bias is critical. To this end, we evaluated OOD generalization using two diverse datasets outside of ImageNet: DTD, a texture dataset  (e.g., stripes, spotted, …),, and CelebA a face dataset with 40 binary attributes (e.g., glasses, wavy hair, …).
>
> Using DTD and CelebA as the validation dataset for our ImageNet trained USAEs show strong evidence of generalization outside of the training distribution (**Table A**). We find consistent activation reconstruction accuracy (measured by MSE and R2), consistent trends in co-firing metrics (**Fig. C**) and visualize some of the most important concepts for these new datasets, along with their associated highest activating images, from ImageNet (**Fig. A and B**). Despite differences in domain and semantics, USAEs trained on ImageNet exhibited robust generalization to both DTD and CelebA. Importantly, many of the concepts identified in these datasets also aligned with high-activation concepts from ImageNet, suggesting that the USAE dictionary captures generalizable structure beyond its training data.
>
> ### **Scalability, automatic strategies beyond visual inspection**
> As in prior work [Ghorbani et al., 2019; Fel et al., 2023c; Kowal et al., 2024b], we determine concept semantics via qualitative inspection: for each concept, we collect its top-activating image examples and generate corresponding spatial token heatmaps to aid interpretation. While this manual approach remains standard, we believe that emerging vision-language models (e.g., vision-capable LLMs) offer promising tools for automating concept summarization and interpretation. We view this as an exciting direction for future work.
>
> ### **USAE vs SAE features**
> We agree with the reviewer that further exploring the differences between USAE concepts and common SAE concepts is an interesting direction. We assume 51dn is asking for a characterization and comparison between universal features mined between independent SAEs vs universal concepts learned from our joint approach. To the best of our knowledge, all previous work does only pairwise analysis to mine overlapping concepts between independent SAEs [Lan et al. 2024], scaling these approaches beyond pairwise comparisons is currently not possible without substantial modifications to the previous methods. Developing a working baseline that extends these post-hoc mining approaches beyond pairwise is out of the scope of our work. We demonstrate that USAEs and independent SAE have fewer overlapping features (USAE-SAE, Sec 4.4 Fig. 7), yet demonstrate much higher overlap between themselves i.e., USAE-USAE, and SAE-SAE (see 11mk Dictionary Stability), further indicating USAEs do learn unique features that are not captured by independent SAEs.
>
> Lan et al. “Sparse Autoencoders Reveal Universal Feature Spaces Across Large Language Models.” ArXiv 2024

---

### Official Review · Reviewer_11mk · 2025-03-17

**Overall Recommendation:** 5

**Summary:**

The paper proposes a recipe to jointly train Sparse AutoEncoders (SAE) across different vision models in a shared (universal) space. The novel idea is to force the SAE to extract features (and, therefore, concepts) that are as shared as possible across models. This shared space enables cross-model and new within-model applications: the former allows studies of how different models encode the same concept, while the latter is used to characterize the distinction between model-specific and shared concept decompositions by comparing SAE and USAE decompositions.

**Claims And Evidence:**

The claims are backed up by strong empirical evidence. The compatibility of extracted features is backed by quantitative measures (sections 4.2 and 4.3), while the interpretability applications are validated through qualitative experiments (sections 4.1 and 4.5). The combination of both makes for a strong case.

**Essential References Not Discussed:**

A few comments on the references:
- The citations to "The Platonic Representation Hypothesis" (PRH) (Huh et al., 2024) aren’t always accurate. The perfect placement is in the related works section under Feature Universality, supporting the discussion. But in the Introduction, it’s cited as a “technique for identifying universal features,” which isn’t quite right. PRH is more of a hypothesis about the existence and the convergence to a shared representation space than a method for discovering them.
- In the Introduction, I would reference "Relative Representations enable Zero-Shot Latent Space Communication" (Moschella et al., 2023). This paper shows that different models, regardless of initialization, pretraining task, or architecture, can exhibit latent spaces that align well enough to be projected into a shared/universal one. That’s very much in line with the idea of universality in USAE-extracted concepts.
- In the "Concept-Based Interpretability" paragraph of the related works section, I would include "Interpreting CLIP’s Image Representation via Text-Based Decomposition" (Gandelsman et al., 2024). This paper analyzes CLIP’s image encoder by decomposing (output-level) image representations into interpretable components (e.g., contributions from individual attention heads), using text representations as a dictionary for the decomposition.

**Experimental Designs Or Analyses:**

The experiments feel carefully designed and informative, with a mix of quantitative and qualitative measures that reinforce the key claims.

My main curiosity in this section is how stable the concept universality metrics (firing entropy, co-firing proportion, and concept consistency) are across different (U)SAE hyperparameters, such as the dictionary size, the chosen models, or the dataset. This would provide a clearer picture of the method’s robustness and how “universal” the extracted features really are. *For example*, does a smaller dictionary recover a subset of the strongest concepts found with a larger dictionary, or does it lead to a more entangled representation? Does the method extract similar concepts with just a subset (2) of the models?

**Methods And Evaluation Criteria:**

The methodology is straightforward (a plus!), with extra credit for the joint optimization process. Instead of optimising every pair, using a single space as a pivot, or adding an extra loss term, the paper takes a cleaner route: individually encoding each model’s activations while jointly decoding across all spaces. However, even though it’s noted in the appendix that the hyperparameters (I guess the SAE ones) require some tuning, it’s unclear how sensitive the method is to these choices. Adding some more information about it would certainly strengthen the paper.

**Other Comments Or Suggestions:**

I'm adding it here because it can be considered a concurrent paper. I think "Sparse Autoencoders Reveal Universal Feature Spaces Across Large Language Models" (Lan et al., 2024) is highly related to this work. In that one, representations of independently learned SAE on LLMs are aligned/compared post-training with similar universality findings.

**Other Strengths And Weaknesses:**

### Extra Strengths
- The writing is **amazingly clear, well-structured, and well-motivated**. The paper is a pleasure to read. I particularly appreciated that many questions that came up during my first read were addressed immediately (e.g., the USAE vs. SAE analysis). The only section that initially gave me trouble was 3.3 (Coordinated Activation Maximization), which could have been improved by adding a high-level sentence or two explaining the methodology in practical terms, making it more accessible.
- Personally, I think cross-model interpretability is a **truly exciting topic**!

### Extra Weaknesses
- **No failure case analysis**. The paper only shows successful examples of extracted concepts, but what happens when the USAE fails? Does it sometimes miss clearly present concepts or identify ones that aren’t? Seeing failure cases where the decomposition disagrees with human intuition would give a better sense of its limitations.
- **Potential bias in the learned dictionary**. Since the USAE extracts concepts from ImageNet (I found this information in the appendix, I would add a sentence in "Implementation details" under section 4 specifying it), it’s likely biased toward the dataset’s structure/composition. But how does the dictionary change if trained on a different dataset? Some analysis of how the discovered dictionary shifts across datasets would be valuable.
- An analysis of **concept stability across runs** is missing. I would expect different runs to produce slightly different dictionaries. But how much is "slightly"? Do the same concepts emerge consistently?

**Questions For Authors:**

I find the paper to be very well-executed, and the core idea is highly interesting. I'm more than open to increasing my rating if the authors could further strengthen it by showing some experiment that addresses the "universality" robustness (see weaknesses 2 and 3 and "Experimental Designs Or Analyses").

To be clear, I don't think this is needed for the paper to be acceptable (that's why I'm already giving a 4), but given how central the “universality” claim is, appearing prominently throughout the text and even in the title, some extra validation would make it more substantiated. With the current validations, I would only refer to it as **a** shared set of concepts to jointly decompose multiple models.

**Relation To Broader Scientific Literature:**

This work has broad implications for the representation learning and interpretability fields. The study of representation compatibility and alignment is an active research topic, and the proposed framework naturally fits within this space by unlocking cross-model studies. One particularly interesting direction would be applying the same method to other modalities (yes, I’m referring specifically to LLMs). A shared concept space is, by definition, abstract. Therefore, it would be interesting to see some cross-domain extensions. A starting point could be applying USAE to the vision branch of CLIP but then applying the learned encoder $\Psi$ and decoder $D$ on the language branch.

**Theoretical Claims:**

N/A

---

> ### Author Rebuttal · Authors · 2025-04-01
>
> We thank 11mk for their thorough analysis of our work. We appreciate that the reviewer found our work to be clearly written and well-motivated. We too believe cross-model interpretability is a truly exciting topic! We answer the questions raised below. For plots and figures, please refer to our temporary anonymous website hosted on Netlify: https://sparkling-queijadas-998747.netlify.app/
>
> ### **Dictionary stability**
> We agree that concept stability is an important metric to consider. Note that our cosine-similarity-based analysis (Sec. 4.4 & Sec. A.3.1) in the main paper measures the stability between independent SAEs and USAEs, which found that USAEs do indeed preserve some of the concepts found in independent SAE dictionaries. However, we appreciate the suggestion to explore stability across training runs of hyperparameter-matching USAEs.
>
> Furthermore, concept stability (via cosine similarity) in SAEs has been analysed in recent work (Fel et. al, 2025). We first independently verify this paper’s findings on ‘individual model’ TopK SAEs, closely matching the stability score they observed (~0.5). We further find that (TopK) USAEs exhibit similar stability to individual TopK SAEs (**Fig. F**), with scores of 0.52, 0.41 and 0.55 for SigLIP, DinoV2, and ViT, resp.
>
> Furthermore, we observe a strong positive correlation between concept stability and importance (**Fig. G**). The most important concepts—those contributing most to reconstruction—are also the most stable across runs, suggesting that universality and stability are linked.
>
>
> ### **Bias in the learned dictionary**
> For discussion on bias in the learned dictionary, see response to 51dn: **Dataset Generalization OOD**.
>
> ### **Comparison: smaller/larger dictionary sizes**
>
> To investigate whether smaller dictionaries favor more universal or entangled concepts, we ablated the expansion factor (4/8/12 = 3072/6144/9216 concepts). We observed that while all dictionary sizes recover many of the same high-importance concepts, **the smallest dictionary (×4) tends to emphasize the most universal and high-entropy concepts (Fig. D and Table B)**, likely due to its limited capacity. In contrast, larger dictionaries better capture low-frequency and low-entropy concepts, which tend to be more model-specific.
>
> ### **Comparison with a subset of models**
> As suggested, we train a USAE on a subset of the models (SigLIP & ViT) and, as expected, find it has a significant overlap of concepts with the 3-model USAE. The stability scores between their dictionaries are 0.47 and 0.5 for SigLIP and ViT, resp. In addition, we again observe the trend of stability/importance correlation: the top 1000 Energy concepts from the 3-model USAE exhibit an average stability of 0.65 with the 2-model USAE. We also note that our firing metrics remain consistent with the 3-model USAE in (**Fig E**).
> These results reinforce our finding that the most **important and universal concepts are also the most stable**, and that universality is recoverable even from subsets.
>
> ### **Universality of concepts**
> We appreciate and agree with your framing: USAEs identify a possible set of shared concepts, not the definitive one. We will revise the paper’s language accordingly.
> Still, we show that the universal concepts we do identify are:
> - **Stable** across training runs,
> - **Important** to activation reconstruction,
> - **Generalizable** across datasets.
>
> These properties form a compelling working definition of universality, and while we cannot claim full coverage, we believe USAEs capture a meaningful and interpretable subset of the shared conceptual space across models.
>
> ### **Failure case analysis**
> We appreciate the suggestion to include failure modes of USAEs. We did find difficult-to-interpret concepts  (concept 2188 in **Fig. H**) or model-biased concepts; e.g., typically related to positional information (concept 5728 in **Fig. H**) which was clearly represented in DinoV2 but not strongly represented in the other models.
>
> While these may appear as failure cases from a universality standpoint, we believe they help illustrate a key strength of USAEs: they surface not only shared concepts, but also **highlight the set of unique concepts of each model**. We will include these findings in the revised version.
>
> ### **Other: References and description refinements**
> We appreciate your feedback regarding Platonic Representations as well as the referral to review and the recommendation to clarify the high-level intuition for Sec. 3.3, and will implement these changes in the revised version.
>
> Thank you again for the detailed review and suggestions for improvement! We hope we have addressed your main concerns, especially regarding the robustness of USAEs. If so, we would appreciate the consideration to raise your score or let us know what we can further demonstrate to help your decision!
>
> Fel et al. “Archetypal SAE: Adaptive and Stable Dictionary Learning for Concept Extraction in Large Vision Models” ArXiv 2025

---

> > ### Comment · Reviewer_11mk · 2025-04-03
> >
> > The extra content is excellent, thank you for your work! I had a look at the other reviews/comments, and I believe the paper is much stronger now, especially with the experiments reinforcing the robustness/universality claims, so I'm pretty confident in raising my score from 4 to 5.

---

> > > ### Author Response · Authors · 2025-04-07
> > >
> > > We thank the reviewer for your time and expertise in providing a comprehensive initial assessment and for taking the time to further examine our rebuttal materials. Furthermore, we greatly appreciate your thoroughness in reviewing not only our responses to your specific comments, but also our responses to other reviewers. We are excited to see that considering the full context of our work with the other reviews and rebuttal materials has strengthened your confidence in our contribution, as reflected in raising your rating from "accept" to "strong accept." We are particularly encouraged that our new results demonstrating USAE stability and robustness resonated with you. The constructive engagement throughout this review process has been invaluable. We strongly believe our work has been improved from your review and will incorporate the feedback in the revised version of the paper.

---

### Decision · Program_Chairs · 2025-05-01

**Decision:**

Accept (poster)

**Comment:**

**Strengths:**

(1) The introduction of USAEs for learning shared concept space across vision models is original and timely.

(2) The paper presents a diverse set of qualitative and quantitative experiments, including cross-model reconstruction, concept alignment, and visualizations.

(3)  The paper is clearly written and well-structured, making complex ideas easily understandable.

**Weaknesses:**

(1) Initial framing of “universality” could be overstated without deeper robustness checks (e.g., stability, OOD generalization, concept overlap).

(2) The distinction between USAEs and post-hoc pairwise SAE comparisons could be better emphasized, especially visually.

(3) Some implementation specifics (e.g., thresholds, interpolation choices) should have been better clarified in the main paper. Additionally, certain metrics (e.g., firing entropy) could benefit from more principled, distribution-aware formulations to strengthen their interpretability and robustness.

**Discussion:**

The paper proposes a sparse autoencoder based framework that opens up new directions in cross-model interpretability. While reviewers 11mk and SLTN reasonably endorsed the work, other reviewers (51dn, 42Tj) expressed mild issues, primarily about the strength of universality claims, implementational clarity. Importantly, the authors responded thoroughly to all criticisms, providing additional experiments on concept stability, OOD generalization, and clarifications around metrics and implementation choices. The rebuttal successfully addressed core concerns of the reviewers and highlighted the authors’ dedication to rigor and openness.

**Recommendation: Accept**

This paper makes a novel and well-supported contribution to the interpretability and representation learning literature. Its insights into shared concept structure across models are timely and practically relevant. While some aspects can be strengthened in future works, the current work is a strong and impactful addition to the field.